# Deciphering multi-way interactions in the human genome

Gabrielle A. Dotson [1,11], Can Chen [2,3,4,11], Stephen Lindsly [1,11], Anthony Cicalo[1], Sam Dilworth[5], Charles Ryan[6,7], Sivakumar Jeyarajan[1], Walter Meixner[1], Cooper Stansbury[1], Joshua Pickard[1], Nicholas Beckloff[8], Amit Surana[9], Max Wicha [10], Lindsey A. Muir[1] & Indika Rajapakse [1,2] ✉

Chromatin architecture, a key regulator of gene expression, can be inferred using chromatin contact data from chromosome conformation capture, or Hi-C. However, classical Hi-C does not preserve multi-way contacts. Here we use long sequencing reads to map genome-wide multi-way contacts and investigate higher order chromatin organization in the human genome. We use hypergraph theory for data representation and analysis, and quantify higher order structures in neonatal fibroblasts, biopsied adult fibroblasts, and B lymphocytes. By integrating multi-way contacts with chromatin accessibility, gene expression, and transcription factor binding, we introduce a data-driven method to identify cell type-specific transcription clusters. We provide transcription factor-mediated functional building blocks for cell identity that serve as a global signature for cell types.

Structural features of the genome are integral to the regulation of gene expression and corresponding generation of cellular phenotypes[1-3]. Aspects of genome structure have been inferred by studying genomic regions that are in close physical proximity. Chromosome conformation capture (3C)-based methods capture these interactions (contacts) through chemical fixation, digestion of DNA, and proximity ligation, followed by sequencing of ligated DNA to identify genomic regions that are in contact. A variety of cell types have now been characterized using Hi-C, a genome-wide 3C-based method, adding substantially to our understanding of genome architecture. However, limitations on read length during sequencing lead to over-representation of simple interactions, predominantly pairwise. Identification of more complex, higher-order interactions can help us build a more complete set of principles of genome architecture.

Multi-way contacts have been identified using targeted 3C-based methods[4-6], through inference from pairwise contacts[7], and on occasion using classical Hi-C[8]. Ligation-free approaches, such as GAM, SPRITE, and ChIA-Drop, have recently enabled large scale capture of multi-way interactions[9-11], though comparisons of different methods find under- and over-representation of higher order contacts in the absence of proximity ligation[12,13].

A recent extension of Hi-C preserves multi-way interactions and uses sequencing of long reads (e.g. Pore-C)[12] to unambiguously identify sets of contacts among multiple loci. Multi-contact 4C sequencing (MC-4C) also uses long-read sequencing to capture contact complexity[14], however, it was designed to capture local topology for individual genes and regulatory regions and does not generate multi-way contacts genome-wide. While direct capture of multi-way contacts can clarify higher order structures in the genome, new frameworks are needed to address unique analysis and representation challenges posed by the multi-way data.

[1]Department of Computational Medicine and Bioinformatics, University of Michigan, Ann Arbor, MI 48109, USA. [2]Department of Mathematics, University of Michigan, Ann Arbor, MI 48109, USA. [3]Department of Electrical Engineering and Computer Science, University of Michigan, Ann Arbor, MI 48109, USA. [4]Channing Division of Network Medicine, Brigham and Women's Hospital and Harvard Medical School, Boston, MA 02115, USA. [5]iReprogram, Ann Arbor, MI 48105, USA. [6]Medical Scientist Training Program, University of Michigan, Ann Arbor, MI 48109, USA. [7]Program in Cellular and Molecular Biology, University of Michigan, Ann Arbor, MI 48109, USA. [8]Oxford Nanopore Technologies, Oxford OX4 4DQ, UK. [9]Raytheon Technologies Research Center, East Hartford, CT 06108, USA. [10]Department of Hematology/Oncology, University of Michigan, Ann Arbor, MI 48109, USA. [11]These authors contributed equally: Gabrielle A. Dotson, Can Chen, Stephen Lindsly. ✉e-mail: indikar@umich.edu

To address this gap, we generated Pore-C data from neonatal and biopsied adult fibroblasts and collected publicly available Pore-C data for B lymphocytes[12] and constructed hypergraphs to represent the multidimensional relationships of multi-way contacts among loci. Hypergraphs are similar to graphs, but hypergraphs contain hyperedges instead of edges. Hyperedges can connect any number of nodes at once, while edges can only connect two nodes[15–17]. Prior work on neural networks highlights the utility of hypergraph representation learning to denoise and analyze existing multi-way contact data and to predict *de novo* multi-way contacts[18]. Here, we use incidence matrix-based representation and analysis of multi-way chromatin structure directly captured by Pore-C data (Algorithm 1), which is mathematically simple and computationally efficient, and yet can provide insights into genome architecture.

In our hypergraph framework, nodes are genomic loci and hyperedges are multi-way contacts among loci. In our incidence matrices, rows are genomic loci and columns are individual hyperedges. This representation enabled quantitative measurements of chromatin architecture through hypergraph entropy and the comparison of different cell types through hypergraph similarity measures. In addition, we integrated Pore-C with other data modalities to discover biologically relevant multi-way interactions, which we term transcription clusters. The cell-type specific transcription clusters we identified support a role in maintaining cell identity, consistent with prior work on transcriptional hubs or factories[19–23]. Furthermore, the formation of transcription clusters in the nucleus is consistent with small world phenomena in networked systems[24,25].

We use the following definitions. Entropy: a measure of structural order in the genome. Hyperedge: an extension of edges where each hyperedge can contain any number of nodes (multi-way contact). Hypergraph: an extension of graphs containing multiple hyperedges. Hypergraph motifs: an extension of network motifs that describe connectivity patterns of 3-way, 4-way, …, $n$-way hyperedges. Incidence matrix: a representation for hypergraphs where rows are nodes and columns are hyperedges. Transcription cluster: a group of genomic loci that colocalize for efficient gene transcription.

## Results

### Capturing multi-way contacts

We conducted Pore-C experiments using human dermal fibroblasts obtained from a skin biopsy and human neonatal dermal fibroblasts, and obtained additional publicly available Pore-C data from B lymphocytes[12]. The experimental protocol for Pore-C is similar to Hi-C, including cross-linking, restriction digestion, and ligation of adjacent ends followed by sequencing (Fig. 1a). Alignment of Pore-C long reads to the genome enables fragment identification and classification of multi-way contacts (Fig. 1b).

Hypergraphs represent multi-way contacts, where individual hyperedges contain at least two loci (Fig. 1c, left). Hypergraphs provide a simple and concise way to depict multi-way contacts and allow for abstract representations of genome structure. Computationally, we represent multi-way contacts as incidence matrices (Fig. 1c, right). For Hi-C data, adjacency matrices are useful for assembly of pairwise genomic contacts. However, since rows and columns represent individual loci, adjacency matrices cannot be used for multi-way contacts in Pore-C data. In contrast, incidence matrices permit more than two loci per contact and provide a clear visualization of multi-way contacts. Multi-way contacts can also be decomposed into pairwise contacts, similar to those in Hi-C, by extracting all pairwise combinations of loci (Fig. 1d).

### Decomposing multi-way contacts

From our Pore-C experiments using adult human dermal fibroblasts, neonatal human dermal fibroblasts, and additional publicly available Pore-C data from B lymphocytes, we constructed

hypergraphs at multiple resolutions (read level, 100 kb, 1 Mb, and 25 Mb)[12].

We first analyzed individual chromosomes at 100 kb resolution by decomposing multi-way contacts into their pairwise contacts. Decomposing Pore-C data into pairwise contacts provides more information than Hi-C, as each Pore-C read can contain many pairwise contacts[12]. It also allows us to identify topologically associated domains (TADs) using established methods[26–28]. We demonstrate identification of TAD boundaries from decomposed multi-way contacts and show intra- and inter-TAD relationships using multi-way contacts (Figs. 2, S1). The loci that frequently participate in these multi-way contacts give rise to the block-like pattern of chromatin interactions often seen in Hi-C data.

### Chromosomes as hypergraphs

To gain a better understanding of genome structure with multi-way contacts, we constructed hypergraphs for entire chromosomes at 1 Mb resolution. We show an incidence matrix of Chromosome 22 as an example in Fig. 3a, and in Fig. 3b, we visualize the distribution of 1 Mb contacts at multiple orders (2-way contacts, 3-way contacts, etc.) on Chromosomes 22. Figure 3c highlights the most common intra-chromosomal multi-way contacts on Chromosome 22 using multi-way contact "motifs", which we use as a simplified way to show hyperedges. Figure 3d shows how multi-way contacts at lower resolutions (25 Mb, 1 Mb) are composed of many multi-way contacts at higher resolutions (100 kb, read level), and Fig. 3e visualizes the multi-way contacts contained in Fig. 3d as a hypergraph.

We also identified multi-way contacts that contain loci from multiple chromosomes. These inter-chromosomal multi-way contacts can be seen at 1 Mb resolution in Fig. 3f and in 25 Mb resolution for both adult fibroblasts and B lymphocytes in Fig. 4. Figure 4 gives a summary of the entire genome's multi-way contacts, by showing the most common intra- and inter-chromosomal multi-way contacts across all chromosomes. We highlight examples of multi-way contacts with loci that are contained within a single chromosome ("intra only"), spread across unique chromosomes ("inter only"), and a mix of both within and between chromosomes ("intra and inter"). We note that many of the "inter only" contacts observed in Fig. 4 may also have intra-chromosomal contacts when viewed at a higher resolution, similar to Fig. 3d. Finally, we found the most common inter-chromosomal multi-way contacts across all chromosomes, which we summarize with five example chromosomes in Fig. 5 using multi-way contact motifs. These multi-way contacts between distant genomic loci may offer insights into the higher-order structural patterning of the genome and its relationship with transcriptional regulation.

### Transcription clusters

We use the following definitions: Transcription cluster: a group of genomic loci that colocalize for efficient gene transcription. Master regulator: a self-regulating transcription factor that regulatory sequences associated with its gene analog. Specialized transcription cluster: a transcription cluster where at least one master regulator binds. Self-sustaining transcription cluster: a transcription cluster where a TF binds and its gene analog is expressed.

Genes are transcribed in short sporadic bursts in areas with high concentrations of transcriptional machinery[29–31], including transcriptionally engaged polymerase and accumulated transcription factors (TFs). Colocalization of multiple genomic loci in these areas could help coordinate or increase efficiency of transcription, an idea supported by studies using fluorescence in situ hybridization (FISH) that show colocalization during active transcription[19]. Simulations also strengthen the idea that genomic loci that are bound by common transcription factors can self-assemble into clusters and form structural patterns commonly observed in Hi-C data[31]. We refer to these instances of highly concentrated areas of transcriptional machinery

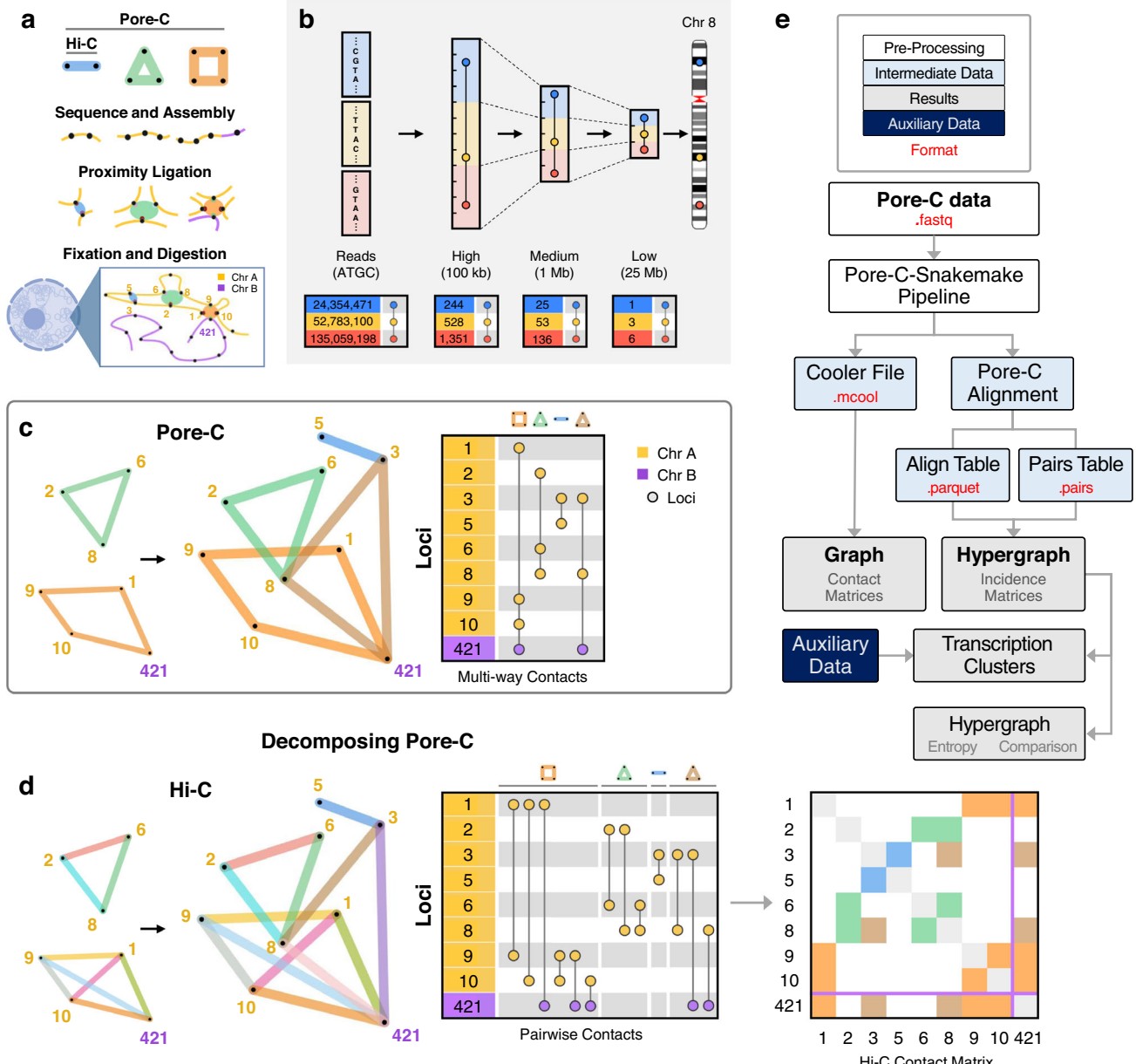

**Fig. 1 | Pore-C experimental and data workflow. a** The Pore-C experimental protocol, which captures pairwise and multi-way contacts (see Methods). **b** Representation of multi-way contacts at different resolutions (top). Incidence matrix visualizations of a representative example from Chromosome 8 in adult human fibroblasts at each resolution (bottom). The numbers in the left columns represent the location of each genomic locus present in a multi-way contact, where values are either the chromosome base-pair position (read-level) or the bin into which the locus was placed (binning at 100 kb, 1 Mb, or 25 Mb). **c** Hypergraph representation of Pore-C contacts (left) and an incidence matrix (right) of four multi-way contacts within (yellow-to-yellow) and between (yellow-to-purple)

chromosomes. Contacts correspond to examples from (**a**). The numbers in the left column represent genomic bins in which a locus resides. Each vertical line represents a multi-way contact, with nodes at participating genomic loci. **d** Multi-way contacts can be decomposed into pairwise contacts. Decomposed multi-way contacts can be represented using graphs (left) or incidence matrices (middle), which when decomposed are interchangeable with traditional Hi-C contact matrices (right). Contacts correspond to examples from (**a**) and (**c**). **e** Flowchart overview of the computational framework. Descriptions of file type formats (red text) are in Supplementary Table 1.

and genomic loci as transcription clusters. The colocalization of multiple genomic loci naturally leads to multi-way contacts, but these interactions cannot be fully captured from the pairwise contacts of Hi-C. Multi-way contacts derived from Pore-C reads can detect interactions between many genomic loci, and are well suited for identifying potential transcription clusters (Fig. 6).

To identify candidate transcription clusters in our Pore-C data, we looked for multi-way contacts with active transcription[32], requiring all loci to be accessible based on ATAC-seq data and at least one locus to have RNA Pol II binding based on ChIP-seq data. Using these criteria,

we identified 12,364, 16,080, and 16,527 potential transcription clusters from neonatal fibroblasts, adult fibroblasts, and B lymphocytes, respectively (Table 1, see Data-driven Identification of Transcription Clusters in Methods). The majority of these clusters involved at least one expressed gene (94.2% in neonatal fibroblasts, 95.0% in adult fibroblasts, 90.5% in B lymphocytes) as well as at least two expressed genes (69.6% in neonatal fibroblasts, 71.9% in adult fibroblasts, 58.7% in B lymphocytes). While investigating the colocalization of expressed genes in transcription clusters, we found that over half of clusters containing multiple expressed genes had common transcription

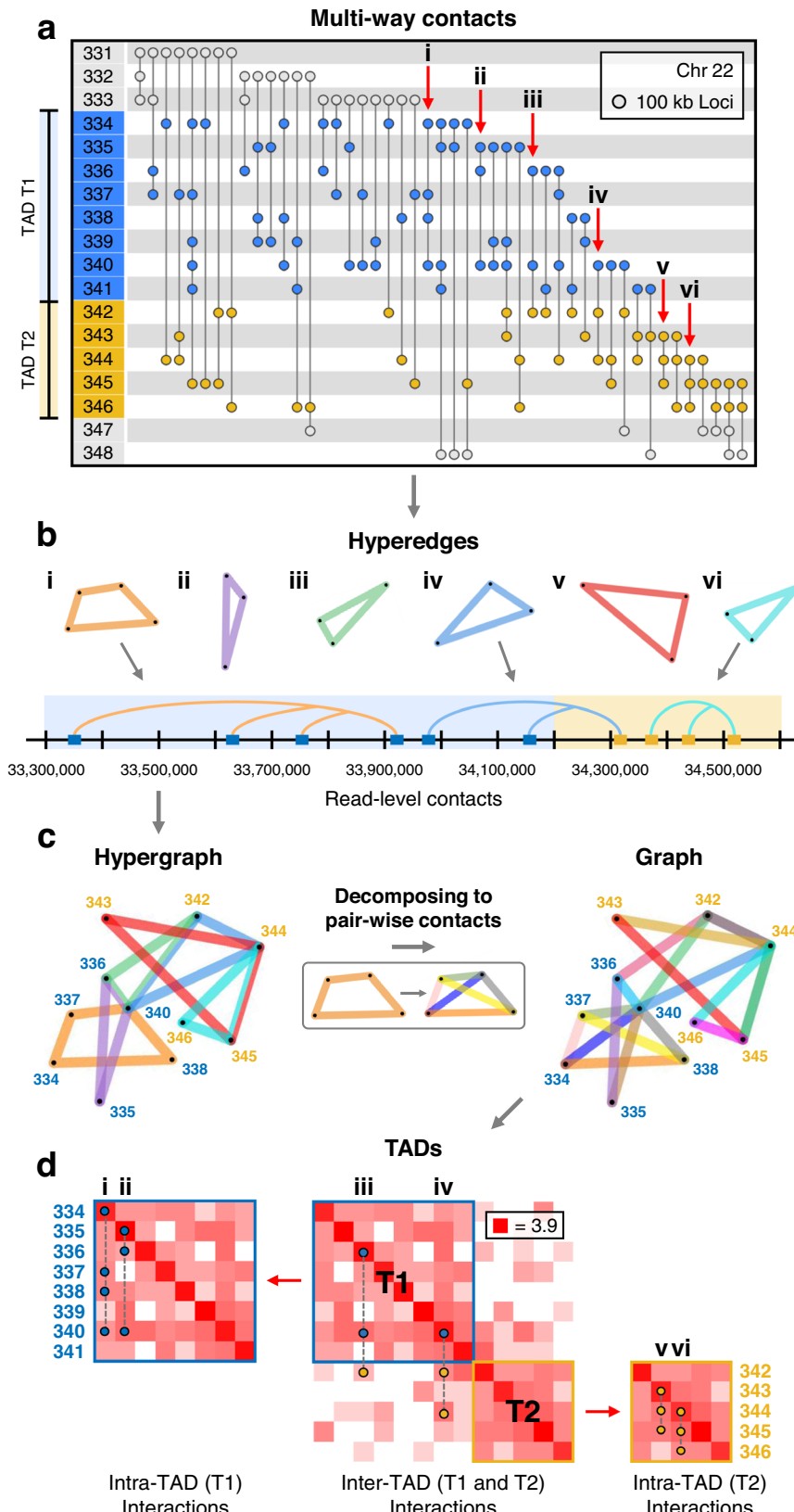

factors based on binding motifs in fibroblasts (61.9% in neonatal fibroblasts, 65.2% in adult fibroblasts) and that over half of these common transcription factors were master regulators (55.9% in neonatal fibroblasts, 63.4% in adult fibroblasts). These proportions were slightly lower in B lymphocytes where we observed that 50.0% of clusters containing multiple expressed genes had common

transcription factors while 46.8% of these common transcription factors were master regulators. Example transcription clusters derived from 3-way, 4-way, and 5-way contacts in fibroblasts and B lymphocytes are shown in Fig. 7. Transcription clusters contained at least two expressed genes with at least one common transcription factor binding motif.

**Fig. 2 | Local organization of the genome. a** Incidence matrix visualization of a region in Chromosome 22 from adult fibroblasts (V1-V4). The numbers in the left column represent genomic loci at 100 kb resolution, vertical lines represent multi-way contacts, where nodes indicate the corresponding locus' participation in this contact. The blue and yellow regions represent two TADs, T1 and T2. The six contacts, denoted by the labels i-vi, are used as examples to show intra- and inter-TAD contacts in (**b**, **c**, and **d**). **b** Hyperedge and read-level visualizations of the multi-way contacts i-vi from the incidence matrix in (**a**). Blue and yellow shaded areas (bottom) indicate which TAD each locus corresponds to. **c** A hypergraph is constructed using the hyperedges from (**b**) (multi-way contacts i-vi from **a**). The hypergraph is decomposed into its pairwise contacts in order to be represented as a graph. **d** Contact frequency matrices were constructed by separating all multi-way contacts within this region of Chromosome 22 into their pairwise combinations. TADs were computed from the pairwise contacts using the methods from[28]. Example multi-way contacts i-vi are superimposed onto the contact frequency matrices. Multi-way contacts in this figure were determined at 100 kb resolution after noise reduction, originally derived from read-level multi-way contacts (see Hypergraph Filtering in Methods).

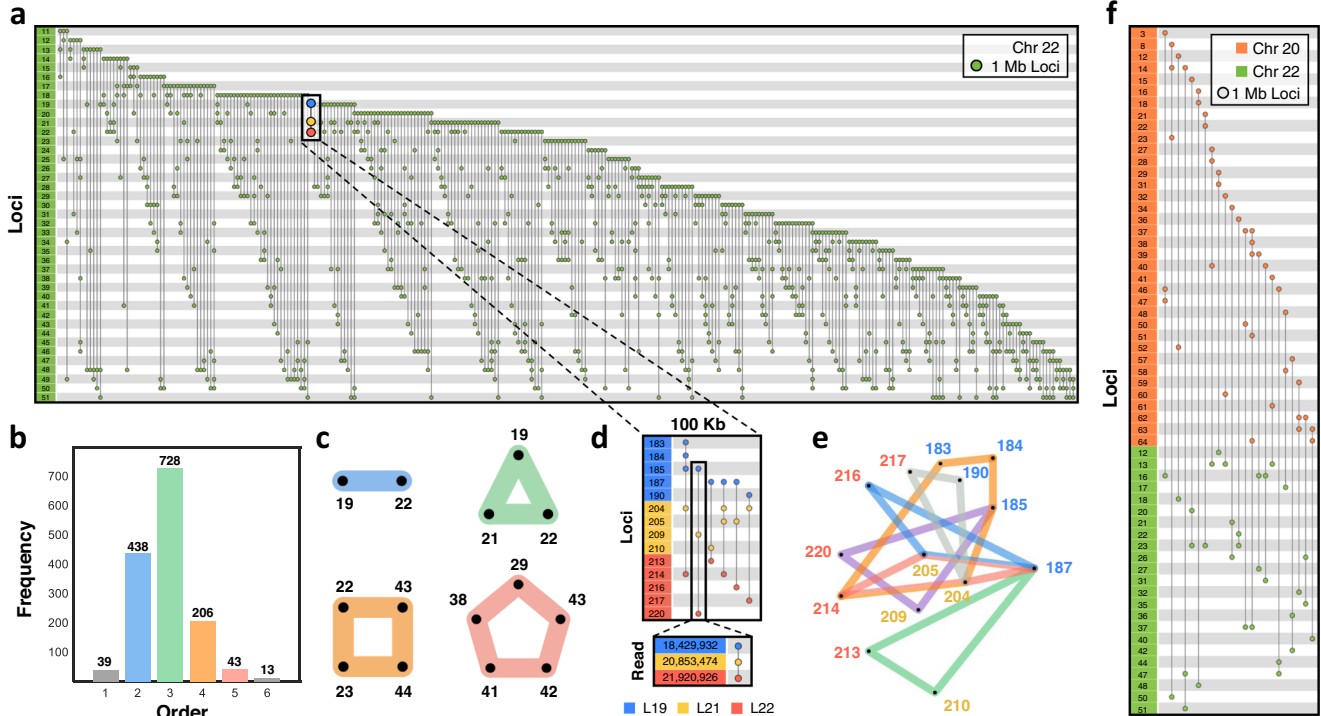

**Fig. 3 | Patterning of intra- and inter-chromosomal contacts. a** Incidence matrix visualization of Chromosome 22 in adult fibroblasts. The numbers in the left column represent genomic loci at 1 Mb resolution. Each vertical line represents a multi-way contact, in which the nodes indicate the corresponding locus' participation in this contact. **b** Frequencies of Pore-C contacts in Chromosome 22. Bars are colored according to the order of contact. Blue, green, orange, and red correspond to 2-way, 3-way, 4-way, and 5-way contacts. **c** The most common 2-way, 3-way, 4-way, and 5-way intra-chromosome contacts within Chromosome 22 are represented as motifs, color-coded similarly to (**b**). **d** Zoomed in incidence matrix visualization in 100 kb resolution shows the multi-way contacts between three 1 Mb loci: L19 (blue), L21 (yellow), and L22 (red). An example 100 kb resolution multi-way contact is zoomed to read-level resolution. **e** Hypergraph representation of the 100 kb multi-way contacts from (**d**). Blue, yellow, and red labels correspond to loci L19, L21, and L22, respectively. **f** Incidence matrix visualization of the inter-chromosomal multi-way contacts between Chromosome 20 (orange) and Chromosome 22 (green) in 1 Mb resolution. Within this figure, all data are from one adult fibroblast sequencing run (V2) and multi-way contacts were determined after noise reduction at 1 Mb or 100 kb resolution accordingly (see Hypergraphs and Hypergraph Filtering in Methods).

We tested the criteria for potential transcription clusters for statistical significance (see Statistics & Reproducibility in Methods). That is, we tested whether the identified transcription clusters are more likely to include genes, and if these genes were more likely to share common transcription factors, than random multi-way contacts. We found that the identified transcription clusters were significantly more likely to include ≥1 gene and ≥2 genes than random multi-way contacts ($p < 0.01$). In addition, transcription clusters containing ≥2 genes were significantly more likely to have transcription factors and master regulators in common ($p < 0.01$). After testing all orders of multi-way transcription clusters together, we also tested the 3-way, 4-way, 5-way, and 6-way (or more) cases individually. We found that all cases were statistically significant ($p < 0.01$) except for clusters with common transcription factors or master regulators in the 6-way (or more) case for both fibroblasts and B lymphocytes. We hypothesize that these cases were not statistically significant due to the large number of loci, naturally leading to an increased overlap with genes. This increases the likelihood that at least two genes will have common transcription

factors or master regulators. Over half of the transcription clusters where the majority of genes contained common transcription factors also contained at least one enhancer in adult fibroblasts (98.2%) and B lymphocytes (87.9%), further suggesting regulatory function within these multi-way contacts[33,34] (Supplementary Table 2). In contrast, only 11.6% of transcription clusters in neonatal fibroblasts exhibited the same properties, which may be a factor of the significantly sparser enhancer annotation available for this cell type compared to the others from the EnhancerAtlas 2.0 database.

To understand how our transcription clusters aligned with factors that have known involvement in chromatin architecture and transcriptional regulation, we evaluated CCCTC-binding factor (CTCF) for binding using ChIP-seq data. CTCF specifically mediates chromatin looping and TAD boundary insulation[35,36] and binds generously throughout the genome[27]. We found significantly higher CTCF binding in our identified transcription clusters compared to multi-way contacts that were not classified as transcription clusters (Supplementary Fig. 3). In adult and neonatal fibroblasts, CTCF binding was nearly two-

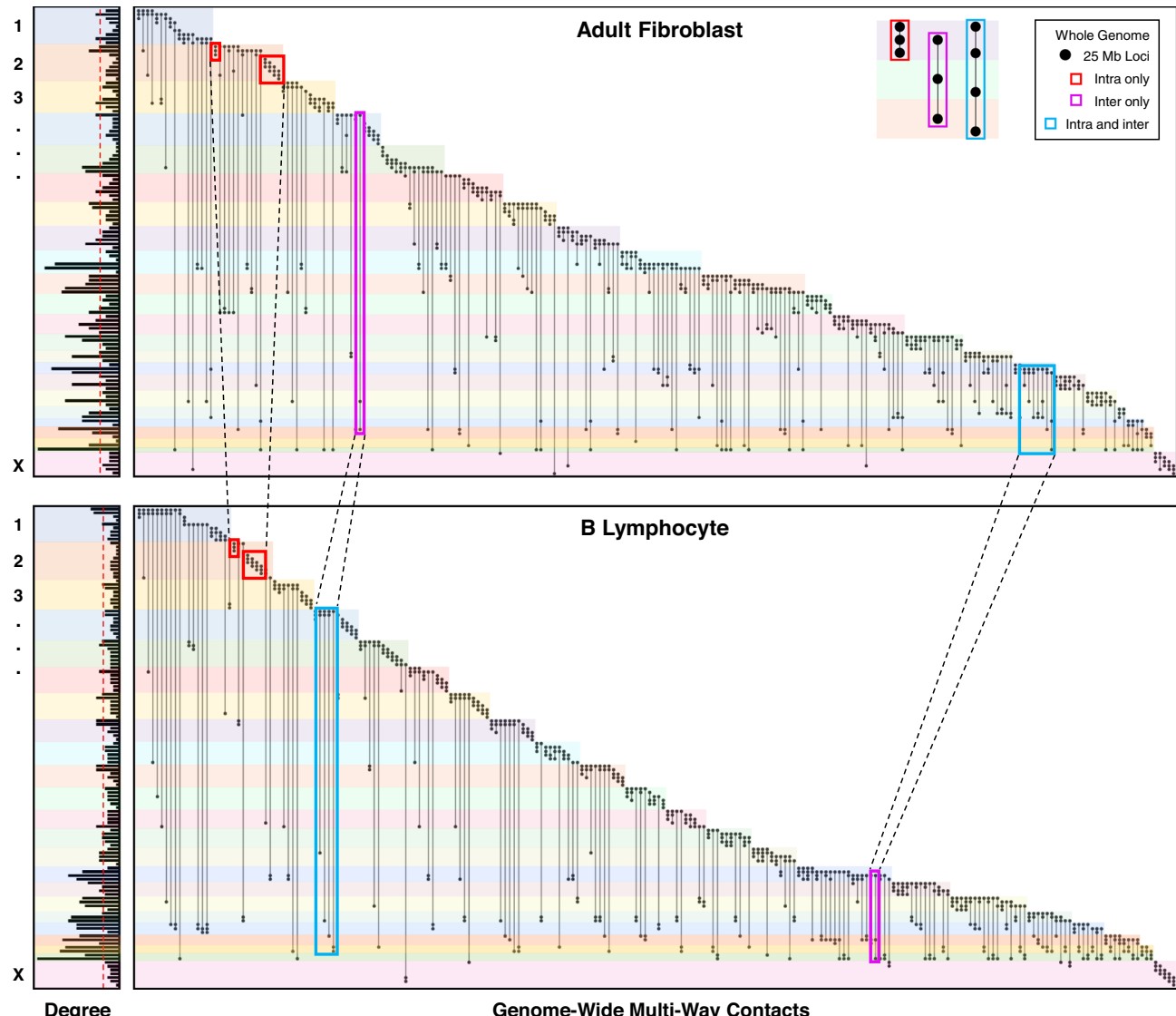

**Fig. 4 | Genome-wide patterning of multi-way contacts.** Incidence matrix visualization of the top 10 most common multi-way contacts per chromosome. Matrices are constructed at 25 Mb resolution for both adult fibroblasts (top, V1-V4) and B lymphocytes (bottom). Specifically, 5 intra-chromosomal and 5 inter-chromosomal multi-way contacts were identified for each chromosome with no repeated contacts. If 5 unique intra-chromosomal multi-way contacts are not possible in a chromosome, they are supplemented with additional inter-chromosomal contacts. Vertical lines represent multi-way contacts, nodes indicate the corresponding locus' participation in a multi-way contact, and color-coded rows delineate chromosomes. Highlighted boxes indicate example intra-chromosomal contacts (red), inter-chromosomal contacts (magenta), and combinations of intra- and inter-chromosomal contacts (blue). Examples for each type of contact are shown in the top right corner. Multi-way contacts of specific regions are compared between cell types by connecting highlighted boxes with black dashed lines, emphasizing similarities and differences between adult fibroblasts and B lymphocytes. Normalized degree of loci participating in the top 10 most common multi-way contacts for each chromosome in adult fibroblast and B lymphocytes are shown on the left. Red dashed lines indicate the mean degree for adult fibroblasts and B lymphocytes (top and bottom, respectively). Genomic loci that do not participate in the top 10 most common multi-way contacts for adult fibroblasts or B lymphocytes were removed from their respective incidence matrices and degree plots. Multi-way contacts were determined at 25 Mb resolution after noise reduction (see Hypergraphs and Hypergraph Filtering in Methods).

fold greater in transcription clusters compared to randomly selected multi-way contacts (80% vs. 45% and 81% vs. 47%). In B lymphocytes, CTCF binding was present at 82% of transcription clusters compared to 59% of random multi-way contacts. We additionally investigated cohesin for its colocalization with CTCF and involvement in regulation of chromatin architecture, such as chromatin loop extrusion[37–40]. In particular, the cohesin subunits RAD21 and SMC3 have been previously linked to CTCF-mediated transcriptional regulation[27]. ChIP-seq data showed preferential binding of RAD21 and SMC3 at transcription clusters compared to random multi-way contacts in adult fibroblasts (79% vs. 42% for RAD21, 71% vs. 37% for SMC3, $p < 0.01$) and B lymphocytes (76% vs. 55% for RAD21, 79% vs. 53% for SMC3, $p < 0.01$)

(Supplementary Fig. 3). Together these data suggest that the identified transcription clusters are important sites of transcriptional regulation, and support a model in which CTCF and cohesin actively mediate multi-way interactions.

We next sought to determine which TFs might be involved in cell type-specific regulation in transcription clusters. For each cell type, we ranked expressed TFs by frequency of binding sites across transcription clusters. Among TFs with the most frequent binding sites, 39% were shared across all three cell types, compared to 72% between adult and neonatal fibroblasts (Supplementary Table 6). Fibroblast and B lymphocyte TF binding sites had less overlap, at 52% (adult) and 45% (neonatal), than binding sites between fibroblasts, supporting cell

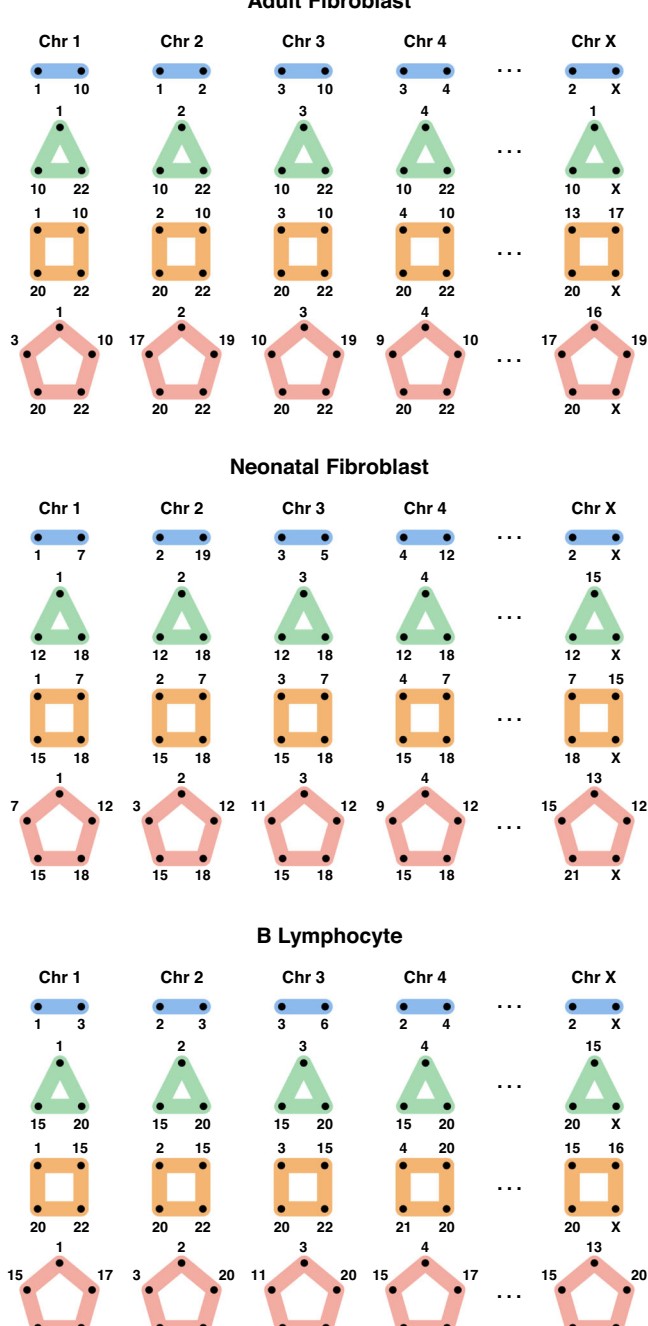

**Fig. 5 | Inter-chromosomal interactions.** The most common 2-way, 3-way, 4-way, and 5-way inter-chromosome combinations for each chromosome are represented using motifs from adult fibroblasts (top), neonatal fibroblasts (center), and B lymphocytes (bottom). Rows represent the combinations of 2-way, 3-way, 4-way, and 5-way inter-chromosomal interactions, and columns are the chromosomes. Inter-chromosomal combinations are determined using 25 Mb resolution multi-way contacts after noise reduction (see Hypergraphs and Hypergraph Filtering in Methods) and are normalized by chromosome length. Here we only consider unique chromosome instances (i.e., multiple loci in a single chromosome are ignored).

type-specific regulation of transcription cluster subsets. Of 18 TFs whose binding sites were unique to the transcription clusters of neonatal fibroblasts, the most frequently occurring was RARB, found at 10.2% of clusters, while in adult fibroblasts, binding sites for ZNF667 were the most frequent among 14 TFs at 6.7% of transcription clusters. In B lymphocytes, binding sites for TFEC were the most frequent

among 161 TFs at 7.6% of transcription clusters. Prior studies support the cell type-specific roles of these uniquely-binding TFs in fibroblasts and B lymphocytes[41,42].

Given the role of TFs in coordinating transcription among clusters of genes[21,43], we hypothesized that TF loci might feature in a subset of transcription clusters. To investigate this question, we looked for the binding motif and encoding gene locus for a given TF within the same transcription cluster, defining this class as a self-sustaining transcription cluster (Fig. 8a, b). We identified nine, eight, and thirteen self-sustaining transcription clusters in adult fibroblasts, neonatal fibroblasts, and B lymphocytes, respectively (Supplementary Table 7). In adult fibroblasts, we observed that the binding motif for FOXO3, a master regulator, exists at a 4-way transcription cluster expressing the *FOXO3* gene. The neonatal fibroblast and B lymphocyte datasets had a self-sustaining transcription cluster in common where STAT3 had a binding motif and the *STAT3* gene was expressed. While self-sustaining transcription clusters demonstrate the capacity for a TF to regulate itself, not every TF co-occupying a transcription cluster with its gene analog is classified as a master regulator (Supplementary Table 7). Therefore, we further stratify these clusters into self-sustaining transcription clusters where the TF is a master regulator and thus binds its gene analog (stronger coupling) and self-sustaining transcription clusters where the TF binds in the cluster but not at is gene analog (weaker coupling). We propose that these strongly-coupled self-sustaining transcription clusters are 'core' transcription clusters that serve as transcriptional signatures for a cell type. It also follows that strongly-coupled self-sustaining transcription clusters are specialized transcription clusters (Supplementary Fig. 4). We then considered two classes of analog-independent transcription clusters - where either a TF and its gene analog occupy different clusters (Fig. 8c) or a TF occupies a cluster, but its gene analog occupies no cluster (Fig. 8d). Since both the TF and gene analog belong to transcription clusters in Fig. 8c, they are coupled, though lesser so than either class of self-sustaining transcription clusters. In contrast, Fig. 8d represents an architecturally uncoupled state–23.3%, 25.7%, and 40.1% of TF gene analogs in adult fibroblasts, neonatal fibroblasts, and B lymphocytes, respectively, were not expressed in any transcription cluster. Lastly, we binned all multi-way contact loci involved in self-sustaining transcription clusters at 100 kb resolution and plotted the interaction frequencies of their decomposed pairwise components (Fig. 8e).

**Algorithm 1**. Multi-way Contact Analysis

1. **Input:** Aligned Pore-C data (**A**), RNA-seq (**R**: gene expression), RNA Pol II (**P**: ChIP-seq), ATAC-seq (**C**: chromatin accessibility), transcription factor binding motifs (**B**)
2. **for** each set of Pore-C data $\mathbf{A}_l \in \mathbf{A}$ **do**
3. Construct incidence matrix $\mathbf{H}_l$ using Algorithm S1
4. Identify transcription clusters $\mathbf{T}_{lp}$, $\mathbf{T}_{lc}$, and $\mathbf{T}_{ls}$ using Algorithm S2
5. Calculate entropy $S_l$ using Algorithm S3
6. **end for**
7. Compute hypergraph distance $d_{ij}$ between pairs $\mathbf{H}_i$ and $\mathbf{H}_j$ with $p \geq 1$ using Algorithm S4
8. Calculate the statistical significance $\alpha_{ij}$ for hypergraph distance $d_{ij}$ using the permutation test in Algorithm S5.
9. **Return:** Hypergraph incidence matrices $\mathbf{H}_l \in \mathbb{R}^{n \times m}$, hypergraph entropy $S_l$, potential transcription clusters $\mathbf{T}_{lp}$, transcription clusters $\mathbf{T}_{lc}$, specialized transcription clusters $\mathbf{T}_{ls}$, and hypergraph distance matrix $[d_{ij}]$ with statistical significance $[\alpha_{ij}]$.

## Discussion

In this work, we introduce a hypergraph framework to study higher-order genome organization from Pore-C long-read sequence data. We demonstrate that higher-order genome architecture can be precisely

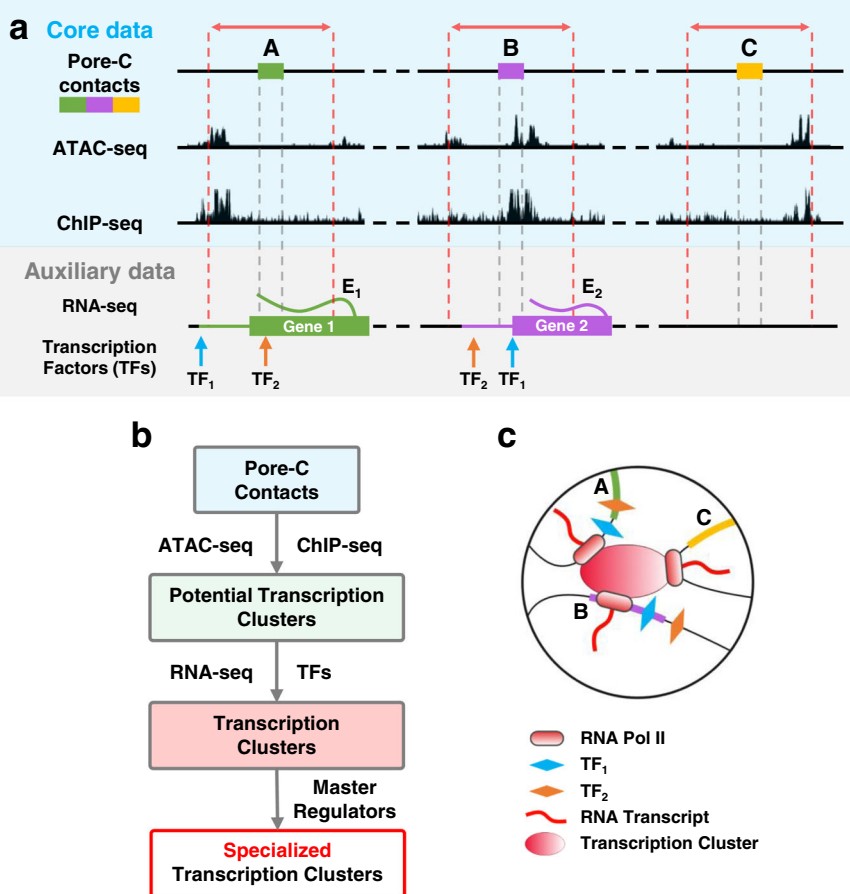

**Fig. 6 | Data-driven identification of transcription clusters. a** Blue shaded area: A 5 kb region before and after each locus in a Pore-C read (region between red dashed lines) is queried for chromatin accessibility and RNA Pol II binding (ATAC-seq and ChIP-seq, respectively). Multi-way contacts between accessible loci that have ≥1 instance of RNA Pol II binding are indicative of potential transcription clusters. Gray shaded area: Gene expression (RNA-seq, $E_1$ for gene 1 and $E_2$ for gene 2, respectively) and transcription factor binding sites ($TF_1$ and $TF_2$) are integrated to determine potential coexpression and coregulation within multi-way contacts with multiple genes. Transcription factor binding sites are queried ±5 kb from the gene's transcription start site (see Data-driven Identification of Transcription Clusters in Methods). Genes are colored based on the overlapping Pore-C locus, and the extended horizontal line from each gene represents the 5 kb flanking region used to query transcription factor binding sites. **b** Pipeline for extracting transcription clusters (Supplementary Methods). **c** Schematic representation of a transcription cluster.

represented and analyzed using hypergraph theory. Using direct capture of multi-way contacts, we identified transcription clusters with physical proximity and coordinated gene expression. Our framework thus enables study of explicit structure-function relationships that are observed directly from data, without needing to infer multi-way contacts. In engineering and social systems, hypergraph representation of data has revealed higher-order organization principles efficiently[15–17,44]. Our work here extends the application of hypergraphs, demonstrating a natural way to represent and analyze genome organization across scales.

Exploring long-range, inter-chromosomal interactions genome-wide offers the opportunity to establish fundamental principles of genome organization. Unbiased capture and study of multi-way contacts can help identify biologically important assemblies that affect transcription, such as transcription clusters[31,45]. This approach can also connect genome organization principles to the study of transcription factors and how they govern cell type-specific network architecture, which resembles small world phenomena[25,46]. Our results support the idea of cell type-specific formation of transcription clusters that serve as a basis for efficient navigation of information within the nucleus, and thereby reflect a signature of small world architecture. Analogous to the behavior of short-path information propagation in social networks, we posit that transcription clusters act as decentralized nodes,

or critical architectures relevant to cell identity. Future work to explore these phenomena systematically will undoubtedly help us understand cell type-specific organization principles. Another exciting direction will be to investigate time series multi-way interactions during cellular transitions such as differentiation and cell reprogramming, with single cell observations. Furthermore, we imagine that multi-way chromatin structure together with spatial transcriptomics will guide us to uncover formation principles in tissue patterning and organogenesis[47,48].

## Methods

### Ethical statement
Primary dermal fibroblasts (IR) were obtained from a punch biopsy from one adult male volunteer (age 42 years) with approval from the Institutional Review Board of the University of Michigan Medical School (HUM00135011) and informed consent. No compensation was provided.

### Cell cultures
Human fibroblasts were maintained in Dulbecco's Modified Eagle Medium (DMEM) supplemented with 10% fetal bovine serum (FBS), 1X Glutamax (Thermo Fisher Scientific Cat no. 35050061) and 1X non-essential amino acid (Thermo Fisher Scientific Cat no. 11140050). BJ

**Table 1 | Summary of multi-way contacts and transcription clusters**

| Order | Multi-way Contacts | Transcription Clusters | Clusters with ≥ 1 Gene | Clusters with ≥ 2 Genes | Clusters with Common TFs | Clusters with Common MRs |
|---|---|---|---|---|---|---|
| 3 | 240,477 | 8,384 | 7,384 | 4,157 | 3,788 | 3,645 |
|   | 301,366 | 8,182 | 7,615 | 5,208 | 4,839 | 4,439 |
|   | 379,165 | 11,261 | 10,581 | 7,518 | 6,890 | 6,778 |
| 4 | 227,352 | 4,345 | 3,972 | 2,686 | 2,435 | 2,341 |
|   | 156,742 | 2,593 | 2,467 | 2,008 | 1,868 | 1,729 |
|   | 181,554 | 3,254 | 3,159 | 2,658 | 2,515 | 2,468 |
| 5 | 196,423 | 1,996 | 1,881 | 1,434 | 1,103 | 957 |
|   | 98,172 | 999 | 976 | 834 | 572 | 443 |
|   | 98,272 | 1,021 | 999 | 877 | 688 | 614 |
| 6+ | 1,000,231 | 1,802 | 1,727 | 1,419 | 932 | 783 |
|   | 178,705 | 590 | 583 | 549 | 368 | 303 |
|   | 142,575 | 544 | 542 | 514 | 395 | 340 |

Multi-way contacts from B lymphocytes (white rows), neonatal fibroblasts (light gray rows), and adult fibroblasts (dark gray rows, V1-V4) are listed after different filtering criteria. Multi-way contacts are considered to be potential transcription clusters if all loci within the multi-way contact are accessible and at least one locus binds RNA Pol II. These multi-way contacts are then queried for nearby expressed genes. If a transcription cluster candidate has at least two expressed genes, we determine whether the majority of these genes have common transcription factors (TFs) through binding motifs. If only two expressed genes are contained within a transcription cluster candidate, we require both genes to have common TFs. From the set of transcription clusters with common TFs, we calculate how many clusters have at least one common master regulator (MR) (Algorithm S2).

## Transcription Clusters

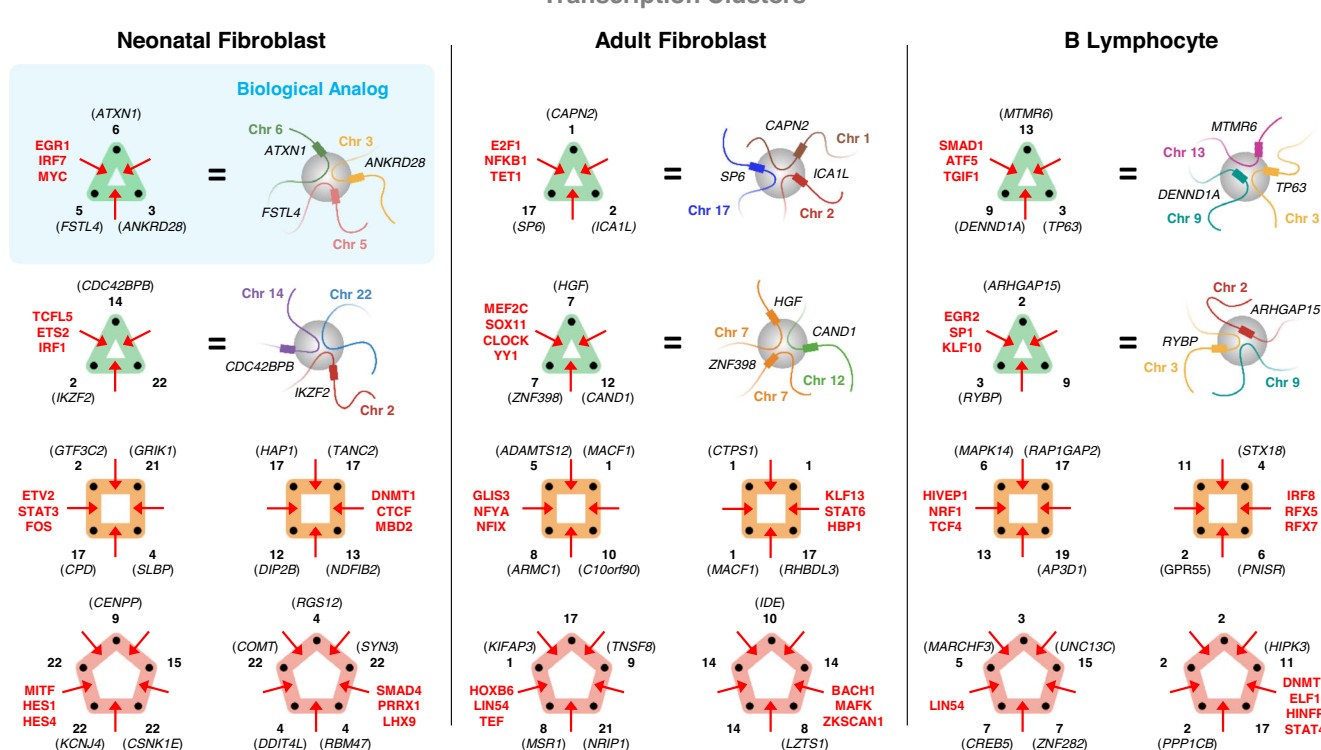

**Fig. 7 | Example transcription clusters.** Six examples of transcription clusters are shown for neonatal fibroblasts (left), adult fibroblasts (center), and B lymphocytes (right) as multi-way contacts (hypergraph motifs). Black labels indicate genes and chromosomes (bold). Red labels correspond to transcription factors shared between the majority of genes within the transcription cluster. For three-way contacts (green motifs), we highlight the transcription clusters' biological analog (blue-shaded box), showing how fragments of chromatin fold and congregate at a common transcription cluster (grey sphere). Each node (black dot) of the

hyperedge and its denoted chromosome and gene in the hypergraph motif corresponds to a single chromatin fragment, colored according to chromosome, in the biological analog. Thus, a three-way hyperedge is depicted by three chromatin fragments in close spatial proximity. Multi-way contacts used for adult and neonatal fibroblasts include all experiments (V1-V4). Examples were selected from the subset of multi-way contacts summarized in the "Clusters with Common TFs" column of Table 1.

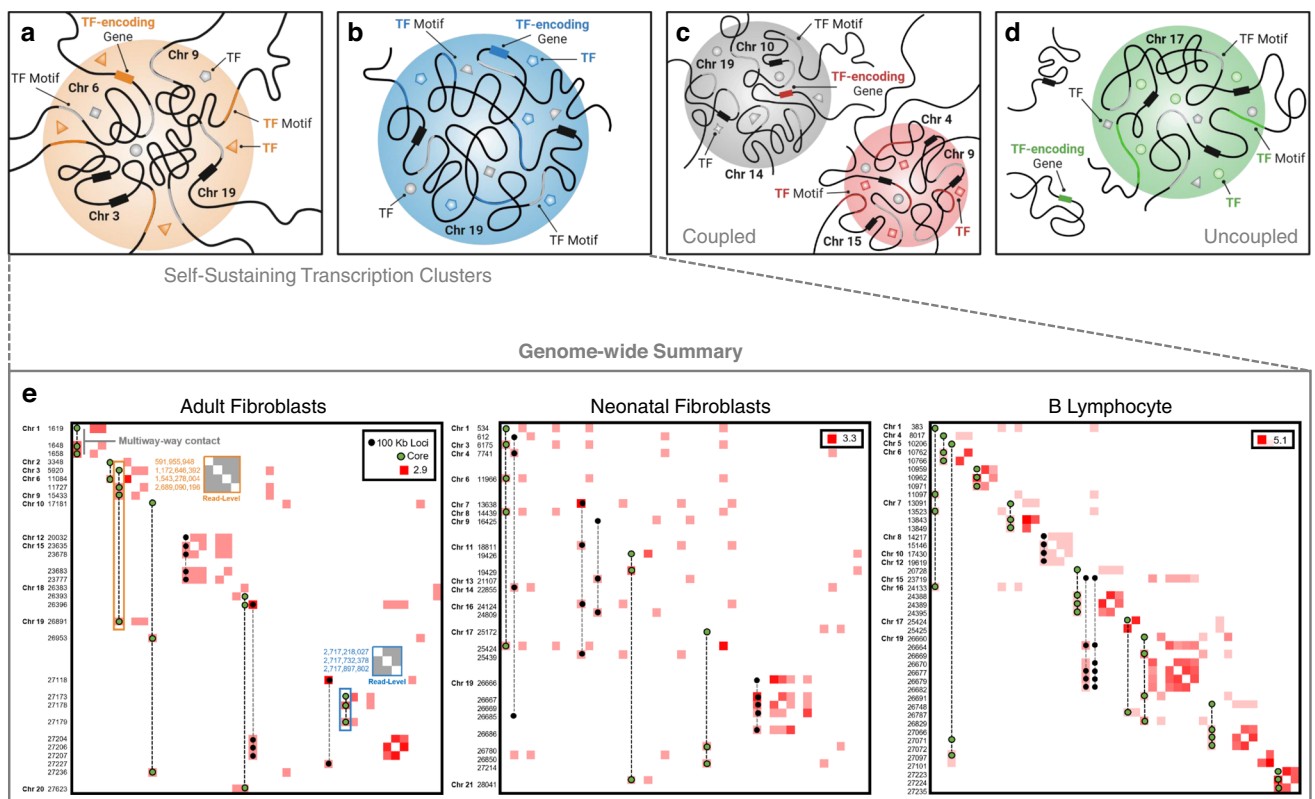

**Fig. 8 | Classes of transcription clusters.** In a self-sustaining transcription cluster, a TF and the gene encoding that TF are both present. The inter- and intra-chromosomal examples in (**a**) and (**b**), respectively, illustrate this phenomenon where in **a** we see the TF of interest (orange triangle) circulating at the cluster, its binding motif present on the chromatin (orange portion), and its corresponding gene expressed (orange rectangle on Chromosome 6). The gray shapes represent additional TFs with binding motifs (gray portion of chromatin) at the cluster. Black rectangles on Chromosomes 3, 9, and 19 represent additional genes present in the cluster. **c** An analog-independent class of transcription clusters where we observe a TF (red square) bind at a transcription cluster (red cluster) and its corresponding gene expressed in a separate transcription cluster (grey cluster), yet not in the same cluster. **d** An analog-independent class of transcription clusters where we observe a TF (green circle) bind at a transcription cluster (green cluster) and its corresponding gene expressed but not within a transcription cluster. **e** Genome-wide cell type-specific self-sustaining transcription clusters extracted from multi-way contact data and decomposed into Hi-C contact matrices at 100 kb resolution. Contact frequencies are log-transformed for better visualization. Frequencies along the diagonal indicate interaction between two or more unique multi-way loci that fall within the same 100 kb bin. Axis labels are non-contiguous 100 kb bin coordinates in chromosomal order. Multi-way contacts that make up the self-sustaining transcription clusters are superimposed. Multi-way contacts with green-colored loci represent 'core' transcription clusters - transcription clusters containing a master regulator and its gene analog. An example read-level contact map for the inter-chromosomal *FOXO3* self-sustaining transcription cluster is denoted by the orange highlighted box in the adult fibroblast contact matrix and a read-level contact map for the intra-chromosomal ZNF320 self-sustaining transcription cluster is denoted by the blue highlighted box. Values along the left axis of these read-level contact matrices are base-pair positions of the contacting loci in the genome.

fibroblasts (RRID:CVCL_3653) were purchased from the American Type Culture Collection (ATCC, Cat no. CRL-2522).

### Cross-linking

Protocols for cross-linking were based on Deshpande et al.[12]. 2.5 million cells were washed three times in chilled 1X phosphate buffered saline (PBS) in a 50 mL centrifuge tube, pelleted by centrifugation at 500 × g for 5 min at 4 °C between each wash. Cells were resuspended in 10 mL room temperature 1X PBS 1% formaldehyde (Fisher Scientific Cat no. BP531-500) by gently pipetting with a wide bore tip, then incubated at room temperature for 10 min. To quench the cross-linking reaction 527 μL of 2.5 M glycine was added to achieve a final concentration of 1% w/v or 125 mM in 10.5 mL. Cells were incubated for 5 min at room temperature followed by 10 min on ice. The cross-linked cells were pelleted by centrifugation at 500 × g for 5 min at 4 °C.

### Restriction enzyme digest

The cell pellet was resuspended in 500 μL of cold permeabilization buffer (10 mM Tris-HCl pH 8.0, 10 mM NaCl, 0.2% IGEPAL CA-630, 100 μL of protease inhibitor cock-tail Roche Cat no. 11836170001) and placed on ice for 15 min. One tablet of protease inhibitor cocktail was dissolved in 1 ml nuclease free water and 100 μL from that was added

to a 500 μL permeabilization buffer. Cells were centrifuged at 500 × g for 10 min at 4 °C after which the supernatant was aspirated and replaced with 200 μL of chilled 1.5X New England Biolabs (NEB) cuts-mart buffer. Cells were centrifuged again at 500 × g for 10 min at 4 °C, then aspirated and re-suspended in 300 μL of chilled 1.5X NEB cuts-mart buffer. To denature the chromatin, 33.5 μL of 1% w/v sodium dodecyl sulfate (SDS, Invitrogen Cat no. 15553-035) was added to the cell suspension and incubated for exactly 10 min at 65 °C with gentle agitation then placed on ice immediately afterwards. To quench the SDS, 37.5 μL of 10% v/v Triton X-100 (Sigma Aldrich Cat no. T8787-250) was added for a final concentration of 1%, followed by incubation for 10 min on ice. Permeabilized cells were then digested with a final concentration of 1 U/μL of NlaIII (NEB-R0125L) and brought to volume with nuclease-free water to achieve a final 1X digestion reaction buffer in 450 μL. Cells were then mixed by gentle inversion. Cell suspensions were incubated in a thermomixer at 37 °C for 18 hours with periodic rotation.

### Proximity ligation and reverse cross-linking

NlaIII restriction digestion was heat inactivated at 65 °C for 20 min. Proximity ligation was set up at room temperature with the addition of the following reagents: 100 μL of 10X T4 DNA ligase buffer (NEB), 10 μL

of 10 mg/mL BSA and 50 μL of T4 Ligase (NEB M0202L) in a total volume of 1000 μL with nuclease-free water. The ligation was cooled to 16 °C and incubated for 6 h with gentle rotation.

### Protein Degradation and DNA Purification
To reverse cross-link, proximity ligated sample was treated with 100 μL Proteinase K (NEB P8107S-800U/ml), 100 μL 10% SDS (Invitrogen Cat no. 15553-035) and 500 μL 20% v/v Tween-20 (Sigma Aldrich Cat no. P1379) in a total volume of 2000 μL with nuclease-free water. The mixture was incubated in a thermal block at 56 °C for 18 hours. In order to purify DNA, the sample was transferred to a 15 mL centrifuge tube, rinsing the original tube with a further 200 μL of nuclease-free water to collect any residual sample, bringing the total sample volume to 2.2 mL. DNA was then purified from the sample using a standard phenol chloroform extraction and ethanol precipitation.

### Nanopore sequencing
Purified DNA was Solid Phase Reversible Immobilization (SPRI) size selected before library preparation with a bead ratio of 0.48X for fragments >1.5 kb. The >1.5 kb products were prepared for sequencing using the protocol provided by Oxford Nanopore Technologies. In brief, 1 $\mu$g of genomic DNA input was used to generate a sequencing library according to the protocol provided for the SQK-LSK109 kit (Oxford Nanopore Technologies, Oxford Science Park, UK, version GDE_9063_v109_revU_14Aug2019). After the DNA repair, end prep, and adapter ligation steps, SPRI select bead suspension (Cat No. B23318, Beckman Coulter Life Sciences, Indianapolis, IN, USA) was used to remove short fragments and free adapters. A bead ratio of 1X was used for DNA repair and end prep while a bead ratio of 0.4X was used for the adapter ligation step. Qubit dsDNA assay (ThermoFisher Scientific, Waltham, MA, USA) was used to quantify DNA and ~300–400 ng of DNA library was loaded onto a GridION flow cell (version R9, Flo-MIN 106D). For adult fibroblasts, 4 sequencing runs were conducted generating a total of 6.25 million reads (referred to as V1-V4). For neonatal fibroblasts, 4 sequencing runs were conducted generating a total of 11.85 million reads.

### Sequence processing
Reads which passed Q-score filtering (`-min_qscore 7`, 4.56 million reads) after base calling on the Oxford Nanopore GridION (Guppy, version 4.0.11) were used as input for the Pore-C-Snakemake pipeline (https://github.com/nanoporetech/Pore-C-Snakemake, commit 6b2f762). The pipeline maps multi-way contacts to a reference genome and stores the hyperedge data in a variety of formats. The reference genome used for mapping was GRCh38.p13 (https://www.ncbi.nlm.nih.gov/assembly/GCF_000001405.39/). Each of the four sequencing runs were assigned a sequencing run label and then concatenated. The combined pipeline outputs were used as standard inputs for all downstream analysis.

### Hypergraphs
A hypergraph is a generalization of a graph. Hypergraphs are composed of hyperedges, which can join any number of nodes[49]. Mathematically, a hypergraph is a pair such that G = {$\mathcal{V},\mathcal{E}$} where $\mathcal{V}$ is the node set and $\mathcal{E}$ is the hyperedge set. Each hyperedge in $\mathcal{E}$ is a subset of $\mathcal{V}$. Examples of hypergraphs include email communication networks, co-authorship networks, film actor/actress networks, and protein–protein interaction networks. For genomic networks, traditional graph-based methods fail to capture contacts that contain more than two genomic loci once, which results in a loss of higher order structural information. Hypergraphs can capture higher order connectivity patterns and represent

multidimensional relationships unambiguously in genomic networks[15,50]. In hypergraphs obtained from Pore-C data, we defined nodes as genomic loci at a particular resolution (e.g. read level, 100 kb, 1 Mb, or 25 Mb bins), and hyperedges as contacts among genomic loci. We switch between these different resolutions both for computational efficiency and visual clarity. Most higher order contacts are unique in Pore-C data at high resolution (read level or 100 kb), so for these data we considered unweighted hypergraphs (i.e. ignore the frequency of contacts). For lower resolutions (1 Mb or 25 Mb), we considered edge weights (frequency of contacts) to find the most common intra- and inter-chromosomal contacts.

### Hypergraph filtering
We performed an additional filtering step while constructing genomic hypergraphs. We first decomposed each multi-way contact into its pairwise combinations at a particular resolution. From these pairwise contacts, we counted the number of times a contact was detected for each pair of loci, and identified the highest frequency locus pairs. Pairwise contacts were kept if they occurred above a certain threshold number, which was set empirically at the 85th percentile of the most frequently occurring locus pairs. For example, in fibroblast data binned at 1 Mb resolution, a locus pair with six detected contacts corresponded to the 85th percentile. Thus all pairs of loci with fewer than six detected contacts were not considered, which increases confidence in the validity of identified multi-way contacts.

### Incidence matrices
An incidence matrix of the genomic hypergraph was an $n$-by-$m$ matrix containing values zero and one. The row size $n$ was the total number of genomic loci, and the column size $m$ was the total number of unique Pore-C contacts (including self-contacts, pairwise contacts, and higher order contacts). Nonzero elements in a column of the incidence matrix indicate genomic loci contained in the corresponding Pore-C contact. Thus, the number of nonzero elements (or column sum) gives the order of the Pore-C contact. The incidence matrix of the genomic hypergraph can be visualized via PAOHvis[51]. In PAOHvis, genomic loci are parallel horizontal bars, while Pore-C contacts are vertical lines that connect multiple loci (see Figs. 1, 2, 3, and 4). Beyond visualization, incidence matrices play a significant role in the mathematical analysis of hypergraphs.

### Data-driven identification of transcription clusters
We used Pore-C data in conjunction with multiple other data sources to identify potential transcription clusters (Fig. 6). Each locus in a Pore-C read, or multi-way contact, was queried for chromatin accessibility and RNA Pol II binding (ATAC-seq and ChIP-seq peaks, respectively). Multi-way contacts were considered to be potential transcription clusters if all loci within the multi-way contact were accessible and at least one locus had binding of RNA Pol II. The loci in potential transcription clusters were then queried for nearby expressed genes. A 5 kb flanking region was added upstream and downstream of each locus when querying for chromatin accessibility, RNA Pol II binding, and nearby genes[52]. Gene expression (RNA-seq) and transcription factor binding site data were then integrated to determine coexpression and coregulation of genes in potential transcription clusters. If a potential transcription cluster candidate had at least two genes present, and these genes had common transcription factors based on binding motifs in the cluster, the potential transcription cluster was determined to be a real transcription cluster. From the set of transcription clusters with common transcription factors, we calculated how many clusters were regulated by at least one master regulator, a transcription factor that also regulates its own gene, and classified these as specialized transcription clusters (Fig. 6).

**Table 2 | Data sources and additional information**

| Data type | Cell type | Description |
|---|---|---|
| Pore-C | IR[a] | Human adult dermal fibroblasts from donor punch biopsy were processed to generate Pore-C data |
| Pore-C | BJ | Human neonatal foreskin fibroblasts (RRID:CVCL_3653) were processed to generate Pore-C data |
| Pore-C | GM12878 | Human B lymphocyte (RRID:CVCL_7526) data were obtained from Deshpande et al.[12] |
| ATAC-seq | IMR-90[a] | Human fetal lung fibroblast (RRID:CVCL_0347) chromatin accessibility data were obtained from ENCODE (ENCFF310UDS)[65] |
| DNase-seq | BJ | Neonatal fibroblast chromatin accessibility data were obtained from ENCODE (ENCFF310UDS) |
| ATAC-seq | GM12878 | B lymphocyte chromatin accessibility data were obtained from ENCODE (ENCFF410XEP) |
| ChIP-seq | IMR-90[a] | Fetal lung fibroblast RNA Polymerase II binding data were obtained from ENCODE (ENCFF676DGR) |
| ChIP-seq | GM12878 | B lymphocyte RNA Polymerase II binding data were obtained from ENCODE (ENCFF912DZY) |
| ChIP-seq | IMR-90 | Fetal lung fibroblast CTCF binding data were obtained from ENCODE (ENCFF203SRF) |
| ChIP-seq | BJ | Neonatal fibroblast CTCF binding data were obtained from ENCODE (ENCFF518RUC) |
| ChIP-seq | GM12878 | B lymphocyte CTCF binding data were obtained from ENCODE (ENCFF951PEM) |
| ChIP-seq | IMR-90[a] | Fetal lung fibroblast RAD21 binding data were obtained from ENCODE (ENCSR000EFJ) |
| ChIP-seq | GM12878 | B lymphocyte RAD21 binding data were obtained from ENCODE (ENCSR000EAC) |
| ChIP-seq | IMR-90[a] | Fetal lung fibroblast SMC3 binding data were obtained from ENCODE (ENCSR000HPG) |
| ChIP-seq | GM12878 | B lymphocyte SMC3 binding data were obtained from ENCODE (ENCSR000DZP) |
| RNA-seq | IMR-90[a] | Fetal lung fibroblast gene expression data were averaged over two samples obtained from ENCODE (ENCFF353SBP, ENCFF496RIW) |
| RNA-seq | IR[a] | Adult fibroblasts were processed to generate gene expression data |
| RNA-seq | BJ | Neonatal fibroblast gene expression data were averaged across two samples obtained from ENCODE (ENCFF477JDG, ENCFF005WBQ) |
| RNA-seq | BJ | Neonatal fibroblasts were processed to generate gene expression data |
| RNA-seq | GM12878 | B lymphocyte gene expression data were averaged across two samples obtained from ENCODE (ENCFF306TLL, ENCFF418FIT) |
| Enhancers | IMR-90[a] | Fetal lung fibroblast enhancer data were obtained from EnhancerAtlas 2.0[66] |
| Enhancers | BJ | Neonatal fibroblast enhancer data were obtained from EnhancerAtlas 2.0[66] |
| Enhancers | GM12878 | B lymphocyte enhancer location data were obtained from EnhancerAtlas 2.0[66] |

[a]indicates data that were combined for identification of transcription clusters in adult fibroblasts.

## Transcription factor binding motifs

Transcription factor binding site motifs were obtained from "The Human Transcription Factors" database[53]. FIMO (https://meme-suite.org/meme/tools/fimo) was used to scan for motifs within ±5 kb of genes' transcription start sites. The results were converted to a 22,083 × 1007 MATLAB table, where rows were genes, columns were transcription factors, and entries were the number of binding sites for a particular transcription factor and gene. The table was then filtered to only include entries with three or more binding sites in downstream computations. This threshold was determined empirically and is adjustable in the MATLAB code.

## Identifying self-sustaining transcription clusters

From identified transcription clusters (Table 1), we obtained a subset containing TF-encoding genes specific to each cell type, yielding 79, 54, and 144 transcription clusters from the adult fibroblast, neonatal fibroblast, and B lymphocyte data, respectively. We then classified these clusters as self-sustaining if the TF binding motif corresponding to the expressed TF-encoding gene was also at the cluster. We further determined whether the self-sustaining TFs were master regulators based on protein-DNA interaction data. Results are summarized in Fig. 8 and Supplementary Table 7.

## Public data sources

Pore-C data for B lymphocytes were downloaded from Deshpande et al.[12]. ATAC-seq and ChIP-seq data were obtained from the Encyclopedia of DNA Elements (ENCODE) to assess chromatin accessibility and RNA Pol II binding, respectively. These data were compared to read-level Pore-C contacts to determine whether colocalizing loci belong to accessible regions of chromatin and had RNA Pol II binding for both fibroblasts and B lymphocytes. RNA-seq data were also obtained from ENCODE to ensure that genes within potential transcription clusters

were expressed in their respective cell types. Additionally, ChIP-seq data for CTCF, RAD21, and SMC3 binding were obtained from ENCODE to evaluate binding preference at transcription clusters. A summary of these data sources is provided in Table 2.

## Hypergraph entropy

Network entropy is often used to measure the connectivity and regularity of a network[17,54,55]. We defined a notion of hypergraph entropy to quantify the organization of chromatin structure from Pore-C data. Denote the incidence matrix of the genomic hypergraph as $\mathbf{H}$. The hypergraph Laplacian matrix is then a $n$-by-$n$ matrix ($n$ is the total number of genomic loci in the hypergraph), which can be computed by

$$\mathbf{L} = \mathbf{D} - \mathbf{H}\mathbf{E}^{-1}\mathbf{H}^{\top} \in \mathbb{R}^{n \times n}, \quad (1)$$

where $\mathbf{D} \in \mathbb{R}^{n \times n}$ is a diagonal matrix containing the degrees of nodes along its diagonal, and $\mathbf{E} \in \mathbb{R}^{m \times m}$ is a diagonal matrix containing the orders of hyperedges along its diagonal. This definition is not equivalent to decomposing each hyperedge into its edge components. This definition also considers the degrees of nodes and hyperedges (i.e., the two degree matrices $\mathbf{D}$ and $\mathbf{E}$ in the equation) of the hypergraph. If we consider a hypergraph with two hyperedges $\{v_1, v_2, v_3\}$ and $\{v_3, v_4\}$, the hypergraph Laplacian matrix and the graph Laplacian matrix (based on decomposing each hyperedge into its edge components) are computed as

$$\mathbf{L} = \begin{bmatrix} \frac{2}{3} & -\frac{1}{3} & -\frac{1}{3} & 0 \\ -\frac{1}{3} & \frac{2}{3} & -\frac{1}{3} & 0 \\ -\frac{1}{3} & -\frac{1}{3} & \frac{5}{3} & -\frac{1}{2} \\ 0 & 0 & -\frac{1}{2} & \frac{1}{2} \end{bmatrix} \text{ and } \mathbf{L}_{\text{graph}} = \begin{bmatrix} 2 & -1 & -1 & 0 \\ -1 & 2 & -1 & 0 \\ -1 & -1 & 3 & -1 \\ 0 & 0 & -1 & 1 \end{bmatrix},$$

respectively. More importantly, the hypergraph Laplacian matrix is well-defined and has useful spectral properties regarding the hypergraph structure[56].

Inspired by von Neumann graph entropy (which utilizes the distribution of the eigenvalues from the graph Laplacian matrix), we define the hypergraph entropy as

$$\text{Hypergraph Entropy} = -\sum_{i=1}^{n} \bar{\lambda}_i \ln \bar{\lambda}_i, \qquad (2)$$

where $\bar{\lambda}_i$ are the normalized eigenvalues of $\mathbf{L}$ such that $\sum_{i=1}^{n} \bar{\lambda}_i = 1$, and the convention $0 \ln 0 = 0$ is used. In mathematics, eigenvalues can quantitatively represent different features of a matrix[57]. Biologically, genomic regions with high entropy are likely associated with high proportions of euchromatin (i.e. less organized folding patterns), as euchromatin is more structurally permissive than heterochromatin[58–60].

We computed the entropy of intra-chromosomal genomic hypergraphs for both fibroblasts and B lymphocytes as shown in Supplementary Table 8. It is expected that larger chromosomes have larger hypergraph entropy because more potential genomic interactions occur in the large chromosomes. However, there are still subtle differences between the fibroblast and B lymphocyte chromosomes, indicating differences in their genome structure. In order to better quantify the structural properties of chromosomes and compare between cell types, it may be useful to introduce normalizations to hypergraph entropy in the future.

## Hypergraph distance
Comparing graphs is a ubiquitous task in data analysis and machine learning[61,61–63]. In order to quantify difference between two genomic hypergraphs $G_1$ and $G_2$ at different scales, we propose to use several hypergraph distance or similarity measures. These measures are based on conversion of hypergraph into a graph representation, see[44] for details. Denote the incidence matrices of two genomic hypergraphs by $\mathbf{H}_1 \in \mathbb{R}^{n \times m_1}$ and $\mathbf{H}_2 \in \mathbb{R}^{n \times m_2}$, respectively. For $i = 1, 2$, construct the adjacency matrices $\mathbf{A}_i$ and normalized Laplacian matrices $\tilde{\mathbf{L}}_i$:

$$\mathbf{A}_i = \mathbf{H}_i \mathbf{E}_i^{-1} \mathbf{H}_i^{\top}, \qquad \tilde{\mathbf{L}}_i = \mathbf{I} - \mathbf{D}_i^{-\frac{1}{2}} \mathbf{H}_i \mathbf{E}_i^{-1} \mathbf{H}_i^{\top} \mathbf{D}_i^{-\frac{1}{2}} \in \mathbb{R}^{n \times n}, \qquad (3)$$

respectively, where $\mathbf{I} \in \mathbb{R}^{n \times n}$ is the identity matrix, $\mathbf{E}_i \in \mathbb{R}^{m_i \times m_i}$ is a diagonal matrix containing the orders of hyperedges along its diagonal, and $\mathbf{D}_i \in \mathbb{R}^{n \times n}$ is a diagonal matrix containing the degrees of nodes along its diagonal[56]. The degree of a node is equal to the number of hyperedges that contain that node. Given these adjacency and normalized Laplacian matrices, we use following three distance measures in our application to determine differences in the two genomic hypergraphs at both local and global scales:

- Hamming Distance: measures local similarity and is based on absolute values of difference between the two adjacency matrices, i.e.,

$$D_H(G_1, G_2) = \frac{1}{n^2} \sum_{j=1}^{n} \sum_{k=1}^{n} |\mathbf{A}_{1,jk} - \mathbf{A}_{2,jk}|,$$

where, the notation $\mathbf{A}_{i,jk}$ implies $jk$-th entry of the matrix $\mathbf{A}_i$.

- Spectral Distance: measures global similarity and is based on the $p$-norm for difference between ordered set of eigenvalues of the two Laplacians, i.e.,

$$D_\lambda(G_1, G_2) = \frac{1}{n} \left( \sum_{j=1}^{n} |\lambda_{1,j} - \lambda_{2,j}|^p \right)^{1/p}, \qquad (4)$$

where $\lambda_{i,j}$ is the $j$th eigenvalue of $\tilde{\mathbf{L}}_i$ for $i = 1, 2$, and $p \geq 1$. In our analysis, we choose $p = 2$.

- DeltaCon Distance: measures both local and global similarity, and is based on the fast belief propagation method of measuring node affinities using the matrix[64], i.e.,

$$\mathbf{S}_i = \left(\mathbf{I} + \epsilon^2 \mathbf{D}_i^a - \epsilon \mathbf{A}_i\right)^{-1},$$

where $0 < \epsilon \ll 1$ is small constant capturing the influence between neighboring nodes, and $\mathbf{D}_i^a$ is the $n \times n$ diagonal matrix with the diagonal entries $\mathbf{D}_{i,jj}^a = \sum_{k=1}^{n} \mathbf{A}_{i,jk}$. DeltaCon then compares the two matrices $\mathbf{S}_1$ and $\mathbf{S}_2$ via the Matusita difference as the measure:

$$D_\Delta(G_1, G_2) = \frac{1}{n^2} \left( \sum_{j=1}^{n} \sum_{k=1}^{n} \left( \mathbf{S}_{1,jk}^{1/2} - \mathbf{S}_{2,jk}^{1/2} \right)^2 \right)^{1/2}, \qquad (5)$$

where we have added a normalization factor $\frac{1}{n^2}$. In our analysis, we found results too insensitive to the choice of $\epsilon$, and report the results for $\epsilon = 10^{-3}$.

Further details on the properties of these different distances can be found in Supplemental Note 1.

We computed hypergraph distance between genome-wide hypergraphs derived from adult fibroblasts, neonatal fibroblasts, and B lymphocytes using the Hamming, spectral, and DeltaCon distances described above and examined distances statistically through a permutation test. Supplementary Fig. 2a1–3 demonstrates that the adult fibroblast and B lymphocyte hypergraphs are significantly different at the chromosome level, especially along Chromosome 21, in stark contrast to the distance between adult and neonatal fibroblasts. Additionally, we computed the same distance measures at the genome level, incorporating inter-chromosomal data, and found that the genomic hypergraphs between fibroblasts and B lymphocytes were significantly different, with a $p$ value of 0 compared to an observed insignificant difference between adult and neonatal fibroblasts ($p$ value of 1) (Supplementary Fig. 2b1–3.)

## Statistics & reproducibility
In order to assess the statistical significance of the transcription cluster candidates we determined using our criteria (Fig. 6), we used a permutation test which builds the shape of the null hypothesis (i.e. the random background distribution) by resampling the observed data over $N$ trials. We randomly selected $n$ 3rd, 4th, 5th, and 6th or more order multi-way contacts from our Pore-C data, where $n$ was based on the number of transcription cluster candidates we determined for each order. For example, we randomly selected $n = 11,261$ multi-way contacts from the set of 3rd order multi-way contacts in fibroblasts (Table 1). For each trial, we determined how many of these randomly sampled "transcription clusters" match our remaining criteria: transcription clusters with ≥1 gene, ≥2 genes, common TFs, and common MRs. The background distribution for each of the criteria was then constructed from these values. The proportion of values in the background distributions that was greater than their counterparts from the data-derived transcription cluster candidates yielded the $p$ value. This analysis was based on the assumption that transcription clusters will be more likely to contain genes and that those genes are more likely to have common transcription factors than random multi-way contacts. For this analysis, we chose $N = 1,000$ trials.

Similarly, we used a permutation test to determine the significance of the measured distances between two hypergraphs.

Suppose that we are comparing two hypergraphs $G_1$ and $G_2$. We first randomly generate $N$ hypergraphs $\{R_i\}_{i=1}^{N}$ that are similar to $G_1$ ("similar" means similar number of node degree and hyperedge size distribution). The background distribution therefore can be constructed by measuring the hypergraph distances between $G_1$ and $R_i$ for $i = 1, 2, ..., N$. The proportion of distances that was greater than the distance between $G_1$ and $G_2$ in this background distribution yielded the $p$ value. For this analysis, we again chose $N = 1000$ trials. See Supplementary Notes for details.

### Reporting summary

Further information on research design is available in the Nature Research Reporting Summary linked to this article.

## Data availability

Source data are provided with this paper. The data discussed in this publication have been deposited in NCBI's Gene Expression Omnibus and are accessible through GEO Series accession number GSE211897. See Table 2 for additional accession codes. Source data are provided with this paper.

## Code availability

All code in our computational framework can be found at: https://github.com/lindsly/Pore-C_Hypergraphs and https://github.com/nanoporetech/pore-c.

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

## Acknowledgements

We thank Professor Peter Cook from the University of Oxford for his guidance on transcription clusters and Professors Gilbert S. Omenn and Alnawaz Rehemtulla for their helpful discussions on figures. We also thank our reviewers for their time and specifically for the suggested example contrasting hypergraph Laplacian matrices and traditional Laplacian matrices, which we adapted and present in Methods. This work is supported in part by the University of Michigan Genome Science Training Program (GSTP) Fellowship funded by NHGRI under Award Number 5T32HG000040-27 to GAD, the Air Force Office of Scientific Research (AFOSR) award FA9550-18-1-0028 to IR and the Defense Advanced Research Projects Agency (DARPA) award 140D6319C0020 and the National Science Foundation (NSF) award 2035827 to iReprogram, LLC.

## Author contributions

I.R. designed research; G.A.D., C.C., S.L., S.D., S.J., W.M., A.C., N.B., C.R., L.M. and I.R. performed research; C.C., S.L., S.D., and I.R. contributed new reagents/analytic tools; G.A.D., C.C., S.L., S.D., C.R., C.S., J.P., A.C., A.S. and I.R. analyzed data; and G.A.D., C.C., S.L., S.D., C.S., N.B., A.S., M.W., L.M., I.R. wrote the paper.

## Competing interests

S.D. is an employee of iReprogram, L.L.C. L.M. and I.R. are co-founders of iReprogram, L.L.C. N.B. is an employee of Oxford Nanopore Technologies. S.L., C.C., and I.R. have submitted a patent application for the computational framework (2115-008250-US-PS1). The remaining authors declare no competing interests.
