## [Peer Review File · Nature Communications]

Reviewers' Comments:

Reviewer #1:

Remarks to the Author:

In this study, "Deciphering Multi-way Interactions in the Human Genome," Lindsly et al create a tool to analyze Pore-C data in a hypergraph framework and extract transcriptional clusters. In addition to testing on publicly available, lymphocyte data, they performed Pore-C experiments in human fibroblasts to demonstrate that utility of their tool to analyze multidimensional genomic architecture.

The manuscript is very organized and straightforward, but few points need more clarification:

- Major Concern: There are already tools that use hypergraphs to analyze multi-dimensional contact data from other methods (i.e. MATCHA Zhang & Ma, Cell Systems 2020). After preprocessing (available pipelines from the original Pore-C makers), why can't these other tools be used. A thorough comparison in detail the advantages of this proposed tool over others.
- The analysis should be extended to reveal biological principles that can be derived from this method. As a few example questions: What is the relationship between gene expression and its ability to participate in multi-way interactions? What is the significance of isolated vs multi-way interacting chromosomes/loci? Which TF binding sites tend to participate in the least vs most multi-way interactions? Why and how does that relate to the expression of genes? Do specific categories of CTCF loops have more multi-way interactions? Much greater depth of analysis is merited to indicate whether or not this method is useful.
- It's not clear how "transcription clusters" differ from A compartments. A comparison between transcription clusters and compartments should be performed. If significant overlap is found, the authors should use standard nomenclature and refer to them as A compartments or multi-way A compartment interactions where applicable. If there is no significant overlap between A compartments and transcription clusters, the authors should explore why.
- The authors generally analyze the data in >100 kb or even >1 Mb bins. Why were these resolutions chosen? It would be useful to evaluate the effect of bin size on the hypergraph analysis. This should include computational considerations (memory and time), multi-way contact frequency, and accuracy of detecting various features.
- It would be useful to provide a more detailed introduction of hypergraphs and how accurately they can describe multidimensional data in a two-dimensional graph.
- In Figure 5, B-lymphocyte group, 4-way contacts (fourth from left): Chromosome 20 appears twice in the same diagram. How can interactions between 20 and 20 be considered interchromosomal?
- Table S2 is not mentioned in the main text.

Reviewer #2:

Remarks to the Author:

This manuscript by Lindsly et al. used Pore-C to study multiway chromatin contacts. The characterization and analysis of multiway interactions are important for studying higher-order genome organization. This work uses the hypergraph/hyperedge concept to analyze multiway contacts, which has been done before by others in the chromatin biology field. However, there are a few major limitations of this work, including the lack of innovation on the computational approach and the lack of concrete biological insights. My major comments are listed below.

1) First and foremost, the authors did not put this work in the context of the existing literature on multiway contacts and transcriptional hubs in the nucleus.

For multiway contact genomic mapping data, the authors should compare their analysis with SPRITE from Mitch Guttman's lab (Quinodoz et al. 2018) which has data in GM12878. SPRITE data addresses the same problem that the authors try to achieve in this work. A direct comparison on the number of multiway contacts and their detailed patterns would be critical to have.

For computational analysis method, the authors need to compare with the methods studying multiway contacts and chromatin hubs from Jian Ma's lab, including MATCHA (Zhang et al. Cell Systems 2020), which develops a hypergraph neural network to improve SPRITE data signals to

identify multiway contacts, and MOCHI (Tian et al. Genome Res 2020), which develops a graph mining approach based on motif clustering to identify transcriptional hubs. These methods are conceptually similar to what the authors are aiming to analyze in this work. For example, if we run MATCHA using Pore-C data, how many multiway contacts can be obtained? How many of these clusters will be consistent with the hubs identified by MOCHI from population Hi-C?

Overall, the authors need to show the advances of the data and analysis methods in this work. The manuscript as it currently stands does not represent sufficient advances both conceptually and technically.

2) The computational workflow described in the paper can be considered a straightforward development of a pipeline (Figure 1). The hypergraph concept is a natural way to characterize multiway contacts. Unfortunately, as a computational work, the paper does not have sufficient depth in problem formulation and algorithm development. For example, the incidence matrix and the hyperedge construction all appear to be rather trivial from a computational method standpoint.

3) The data analysis of multiway contacts from the Pore-C data is inadequate as a research paper. In multiple figures, the authors presented examples from the hyperedge identification (multiway contacts) and the identified patterns of hyperedges. However, there is no rigorous and careful analysis of these multiway contacts to connect to both mechanisms and functions of nuclear organization and other 3D genome features. The authors need to significantly re-design this part of the manuscript by comparing to prior work including the SPRITE data and related computational approaches in the literature. What do these identified multiway contacts implicate? How do we know they are solid and reliable? What's the connection to other 3D genome properties? What's the difference compared to grouping pairwise contacts inferred from Hi-C? What type of noise is there in Pore-C? All these questions are unknown from the current work.

4) The analysis on transcription clusters is of potential interest. However, the data is dramatically underutilized with only a few examples presented. This part of the work should be redesigned to 1) compare with prior methods to identify transcriptional hubs and 2) analyze the functional significance of the identified clusters by comparing to orthogonal datasets, rather than simply listing the number of clusters, the number of genes and a few examples (Figure 7, Table 1).

Response to Reviewer Comments on Manuscript NCOMMS-21-35046-T

Title: Deciphering Multi-way Interactions in the Human Genome

We thank the editor and the Reviewers for their time and valuable comments, and here we address the issues raised by the reviewers. Please see the following reviewer comments (in black) and our point-by-point responses (in blue). Based on the reviewers' suggestions, we have revised our manuscript significantly, improved the clarity of our findings, and addressed errors throughout the text.

Comments by Reviewer 1

1. There are already tools that use hypergraphs to analyze multi-dimensional contact data from other methods (i.e. MATCHA Zhang & Ma, Cell Systems 2020). After preprocessing (available pipelines from the original Pore-C makers), why can't these other tools be used. A thorough comparison in detail the advantages of this proposed tool over others.

We thank the Reviewer for their comments. It is our understanding that the MATCHA algorithm has two main uses: predicting *de novo* multi-way contacts and denoising multi-way contact data. In terms of predicting *de novo* multi-way contacts, the authors state that MATCHA is only applied to ChIA-Drop data. This is because ChIA-Drop "only contains intra-chromosomal hyperedges and has a maximum 1D genomic distance for interactions..." while SPRITE does not have these same constraints [1, 2]. The authors of MATCHA appear to believe that their algorithm cannot be applied to SPRITE, because it would not be practical to enumerate all possible triplet combinations genome-wide [1]. This exact reasoning would apply to Pore-C data as well.

The denoising aspect of MATCHA decomposes all triplets in the training data into pairwise edges, calculates the average values for intra-chromosomal and inter-chromosomal interactions, and then uses these values to binarize the Hi-C contact matrix. Predicted probability scores were then calculated for the triplets, and the authors found that triplets with high probability scores were likely to have all three pair-wise interactions supported by the Hi-C edges. Our method of denoising is somewhat similar, since we are decomposing all multi-way contacts into their pairwise interactions and removing any pairwise contacts that occur fewer times than a threshold. These interactions are then removed from the corresponding multi-way contact data, ensuring that the multi-way contacts we observe are from loci that often contact one another. Instead of establishing a probability score for each multi-way contact, we are effectively removing specific loci from each multi-way contact that rarely interact with the rest of the loci in that multi-way contact. While this inevitably reduces the size of the multi-way contacts, it increases confidence that the preserved multi-way interactions between those loci are real.

Finally, the MATCHA algorithm was designed for and tested on SPRITE and ChIA-Drop data but has not been validated on Pore-C data. It is outside of the scope of our manuscript to adapt a computational algorithm to fit a new data type, but we would be very interested in using this algorithm if it was extended to other sources of data. We have added a reference to the manuscript by Zhang *et al.* and a short discussion of MATCHA starting at line 25 to clarify these points.

2. The analysis should be extended to reveal biological principles that can be derived from this method. As a few example questions: What is the relationship between gene expression and its ability to participate in multi-way interactions? What is the significance of isolated vs multi-way interacting chromosomes/loci? Which TF binding sites tend to participate in the least vs most multi-way interactions? Why and how does that relate to the expression of genes? Do specific categories of CTCF loops have more multi-way interactions? Much greater depth of analysis is merited to indicate whether or not this method is useful.

We thank the Reviewer for their input. Gene expression does not necessitate a multi-way contact and multi-way contacts do not need to contain expressed genes to exist. We find that given certain criteria for a multi-way contact (i.e., open chromatin and binding of RNA Pol II), the vast majority of multi-way contacts contain at least one expressed gene, if not multiple. We then explore the biological significance of transcription clusters with multiple expressed genes by evaluating how many of these clusters' genes have transcription factors in common and have added some biologically relevant examples in Figure 7. Regarding CTCF binding, our data indicates that transcription clusters are significantly more highly enriched for CTCF binding than multi-way contacts that are not classified as transcription clusters. CTCF binding at transcription clusters was nearly double the CTCF binding observed among other multi-way contacts. Please see the addition beginning at line 129 and the new Supplementary Figure S4. Also, we see particular TFs that frequently bind in fibroblasts and not B-lymphocytes, which gives rise to certain genes being expressed in only fibroblasts. Conversely, we see particular TFs that frequently bind in B-lymphocytes and not fibroblasts, which gives rise to gene expression specific to B-lymphocytes. We have added Table S6 that highlights the most frequently occurring TF binding sites identified in transcription clusters.

Figure 1: This figure is borrowed from the manuscript (Figure 6). It highlights the criteria necessary to classify a multi-way contact as a potential transcription cluster. We then evaluate these multi-way contacts to see if they contain expressed genes.

- It's not clear how "transcription clusters" differ from A compartments. A comparison between transcription clusters and compartments should be performed. If significant overlap is found, the authors should use standard nomenclature and refer to them as A compartments or multi-way A compartment interactions where applicable. If there is no significant overlap between A compartments and transcription clusters, the authors should explore why.

We thank the Reviewer for their comments. There is distinct difference between A compartments and transcription clusters. A compartments refer to generally euchromatic regions of the genome where chromatin is accessible and available for active transcription. For a multi-way contact to be classified as a transcription cluster, we require that all loci involved fall in an accessible chromatin region. It follows then that transcription clusters will form with loci in A compartments of the genome. However, not all loci in A compartments participate in transcription clusters. In order to be considered a transcription cluster, the loci must also contain an RNA Pol II binding site, express or contact at least two other loci that express a gene, and contain a common TF binding site in the majority of the involved genes. Thus, the notion of a transcription cluster extends beyond the bilateral condition of A/B compartment classification.

- The authors generally analyze the data in >100 kb or even >1 Mb bins. Why were these resolutions chosen? It would be useful to evaluate the effect of bin size on the hypergraph analysis. This should include computational considerations (memory and time), multi-way contact frequency, and accuracy of detecting various features.

We thank the Reviewer for their input. We have used 25 Mb, 1 Mb, 100 kb, and read level resolutions throughout our manuscript. We use certain resolutions to establish relationships genome-wide (25 Mb), between individual chromosomes (1 Mb), in local features like TADs (100 kb), and for very specific regions in our transcription cluster analysis (read level). We also show how "individual" low resolution interactions can contain many higher order multi-way contacts (e.g., Figure 3A and Figure 3D). We use various resolutions both for computational efficiency and visual clarity, but we understand that we transition between these resolutions multiple times without explicitly describing our reasoning. In response, we have added a clarifying statement to the methods section titled 'Hypergraphs' starting on line 257.

- It would be useful to provide a more detailed introduction of hypergraphs and how accurately they can describe multidimensional data in a two-dimensional graph.

We thank the Reviewer for this suggestion. We have added additional explanation of hypergraphs and how they can be used to represent Pore-C data starting on line 250 of our revised manuscript under the methods section titled 'Hypergraphs'.

- In Figure 5, B-lymphocyte group, 4-way contacts (fourth from left): Chromosome 20 appears twice in the same diagram. How can interactions between 20 and 20 be considered interchromosomal?

We thank the Reviewer for pointing out this error. We have resolved this in the updated Figure 5.

7. Table S2 is not mentioned in the main text.

We thank the Reviewer for their feedback. We have added text on line 126 that addresses Table S2.

Comments by Reviewer 2

1. First and foremost, the authors did not put this work in the context of the existing literature on multiway contacts and transcriptional hubs in the nucleus. For multiway contact genomic mapping data, the authors should compare their analysis with SPRITE from Mitch Guttman's lab (Quinodoz et al. 2018) which has data in GM12878. SPRITE data addresses the same problem that the authors try to achieve in this work. A direct comparison on the number of multiway contacts and their detailed patterns would be critical to have. For computational analysis method, the authors need to compare with the methods studying multiway contacts and chromatin hubs from Jian Ma's lab, including MATCHA (Zhang et al. Cell Systems 2020), which develops a hypergraph neural network to improve SPRITE data signals to identify multiway contacts, and MOCHI (Tian et al. Genome Res 2020), which develops a graph mining approach based on motif clustering to identify transcriptional hubs. These methods are conceptually similar to what the authors are aiming to analyze in this work. For example, if we run MATCHA using Pore-C data, how many multiway contacts can be obtain? How many of these clusters will be consistent with the hubs identified by MOCHI from population Hi-C? Overall, the authors need to show the advances of the data and analysis methods in this work. The manuscript as it currently stands does not represent sufficient advances both conceptually and technically.

We thank the Reviewer for their comments. We agree that MOCHI is an important tool that identifies heterogeneous interactome modules (HIMs) [3]. The MOCHI algorithm determines HIMs by finding groups of gene loci that are close together in space more frequently than expected, through Hi-C data, and are regulated by common transcription factors. This is fundamentally different than our approach, since we report direct observations of multi-way contacts that contain gene loci. We believe that an in-depth comparison between methods to identify transcription clusters or HIMs would be better suited for a different manuscript, since here we are focusing on providing a straightforward yet sound method to process and analyze multi-way contacts that can be used to make insights into genome architecture and gene transcription. We offer an algorithm for identifying transcription clusters as one possible implementation of this data, but multi-way contacts from Pore-C are not limited to just this application. Regarding MATCHA and SPRITE, we refer the Reviewer to our detailed response to Reviewer 1's first question.

2. The computational workflow described in the paper can be considered a straightforward development of a pipeline (Figure 1). The hypergraph concept is a natural way to characterize multiway contacts. Unfortunately, as a computational work, the paper does not have sufficient depth in problem formulation and algorithm development. For example, the incidence matrix and the hyperedge construction all appear to be rather trivial from a computational method standpoint.

We thank the Reviewer for their input. The computational pipeline presented in our manuscript is meant to be easily accessible and interpretable. We offer a concise workflow to process and analyze multi-way contacts without the high computational complexity of other algorithms. While the Reviewer suggests that this is a detriment to our work, we believe that the so-called "trivial" construction of hyperedges from Pore-C data is a major strength of our manuscript.

3. The data analysis of multiway contacts from the Pore-C data is inadequate as a research paper. In multiple figures, the authors presented examples from the hyperedge identification (multiway contacts) and the identified patterns of hyperedges. However, there is no rigorous and careful analysis of these multiway contacts to connect to both mechanisms and functions of nuclear organization and other 3D genome features. The authors need to significantly re-design this part of the manuscript by comparing to prior work including the SPRITE data and related computational approaches in the literature. What do these identified multiway contacts implicate? How do we know they are solid and reliable? What's the connection to other 3D genome properties? What's the difference compared to grouping pairwise contacts inferred from Hi-C? What type of noise is there in Pore-C? All these questions are unknown from the current work.

We thank the Reviewer for their comments. In the revised manuscript, we have expanded our analysis to improve clarity on the soundness of our method and provide more context for it. In summary, our revised manuscript includes: A simple representation of multi-way contacts from Pore-C data, identification of transcription clusters (analogous to transcription factories or transcriptional hubs), comparisons between the transcription clusters between different cell types, and inclusion of additional genomic data (CTCF and cohesin subunit binding) [4]. We have revised our manuscript to include the identification and comparison of transcription clusters in three different cell types. We included further context and discussion of these data to highlight biological implications and expanded the notion of transcription clusters to include self-regulating transcription factors [5]. We have explored this phenomenon genome-wide. We also include discussion of how to compare different cell types quantitatively using Pore-C data (please see below for our technical paper that directly addresses this [6]). Finally, we have added an acknowledgement of the contribution of SPRITE with appropriate citations on line 16 and emphasized

the importance of using multi-way contacts to better understand the relationship between genome structure and gene expression.

4. The analysis on transcription clusters is of potential interest. However, the data is dramatically underutilized with only a few examples presented. This part of the work should be redesigned to 1) compare with prior methods to identify transcriptional hubs and 2) analyze the functional significance of the identified clusters by comparing to orthogonal datasets, rather than simply listing the number of clusters, the number of genes and a few examples (Figure 7, Table 1).

We thank the Reviewer for their suggestion on how we can expand the analysis of our identified transcription clusters. Regarding the Reviewer's first point, we acknowledge that prior methods have identified transcriptional hubs (analogous to transcription clusters) and provide functional insights. However, transcriptional hubs identified by SPRITE, for example, focused more on characterizing where hubs of DNA localize relative to nuclear bodies like the nucleolus and nuclear speckles, rather than exploring the functional significance of the wider breadth of multi-way contacts as we did. The one set of hubs they do highlight contain a cluster of histone genes. From our Pore-C dataset using the same B lymphocyte cell line SPRITE used, we too were able to identify histone gene-containing clusters - 20 of our transcription clusters contained histone genes, 8 of which contained more than one histone gene (results not included in the manuscript). Regarding the Reviewer's second point, we now provide a more thorough functional exploration of transcription clusters, looking at CTCF and cohesin subunit (RAD21 and SMC3) binding preference at transcription clusters (new Figure S4), uniquely binding TFs at transcription clusters across cell types, and coupling behavior across transcription clusters (new Figure 8). Our findings are summarized starting at line 129 in the manuscript. Briefly, we find that transcription clusters contain more instances of CTCF, RAD21, and SMC3 binding than multi-way contacts that were not classified as transcription clusters, and a number of cell-type specific TFs belong to self-sustaining transcription clusters, where a TF and its gene analog are in the same cluster.

References

- [1] Ruochi Zhang and Jian Ma. Matcha: Probing multi-way chromatin interaction with hypergraph representation learning. *Cell systems*, 10(5):397–407, 2020.
- [2] Sofia A. Quinodoz, Noah Ollikainen, Barbara Tabak, Ali Palla, Jan Marten Schmidt, Elizabeth Detmar, Mason M. Lai, Alexander A. Shishkin, Prashant Bhat, Yodai Takei, Vickie Trinh, Erik Aznauryan, Pamela Russell, Christine Cheng, Marko Jovanovic, Amy Chow, Long Cai, Patrick McDonel, Manuel Garber, and Mitchell Guttman. Higher-Order Inter-chromosomal Hubs Shape 3D Genome Organization in the Nucleus. *Cell*, 0(0):1–14, 2018.
- [3] Dechao Tian, Ruochi Zhang, Yang Zhang, Xiaopeng Zhu, and Jian Ma. Mochi enables discovery of heterogeneous interactome modules in 3d nucleome. *Genome research*, 30(2):227–238, 2020.
- [4] Peter R Cook and Davide Marenduzzo. Transcription-driven genome organization: a model for chromosome structure and the regulation of gene expression tested through simulations. *Nucleic Acids Research*, (16):1–12, 2018.
- [5] Ralph Stadhouders, Guillaume J Filion, and Thomas Graf. Transcription factors and 3d genome conformation in cell-fate decisions. *Nature*, 569(7756):345–354, 2019.
- [6] Amit Surana, Can Chen, and Indika Rajapakse. Hypergraph similarity measures. *arXiv preprint arXiv:2106.08206*, 2021.

Reviewers' Comments:

Reviewer #1:

Remarks to the Author:

I'm satisfied with the authors' responses.

Reviewer #2:

Remarks to the Author:

The authors did not make an effort to carefully address the comments from this reviewer. The main problems remain: 1) The analytic approaches used in this work lack novelty and rigor; and some methods are technically flawed, e.g., hypergraph entropy and hypergraph distance; 2) There is a major lack of detailed comparisons with prior work and data; 3) The biological analysis is inadequate as a research paper. Based on the responses to review comments and the revised manuscript, the authors appear to have incorrect and insufficient understanding of the existing work in chromatin biology and computational techniques. Overall, this work as it currently stands does not have sufficient conceptual advances in the field of 3D chromatin organization. In particular, the computational methods lack technical novelty and rigor and the manuscript still has not demonstrated that the methods have theoretical and practical evidence to produce reliable results. My main comments are listed below.

1. It is disappointing that the authors have not provided sufficient comparisons with existing work and available data as I previously pointed out. The authors' responses to the review also incorrectly represented prior work.

The authors mentioned that "Recent methods for analyzing Hi-C data have inferred multi-way interactions from pairwise interactions[4-9], though with the limitation of some ambiguity inherent to artificial assembly of pairwise ligation junctions or machine learning-based predictions." and cited not only the computational methods that infer multi-way interactions from pairwise interactions (Olivares-Chauvet et al.[4] and Liu et al. [8]) but also include new experimental assays that are ligation free and capture multiway interactions directly (such as SPRITE: Quinodoz et al., ChIA-Drop: Zheng et al., a review paper: Kempfer et al., protocol for SPRITE: Quinodoz et al.). This clearly is a misrepresentation of the existing work in the literature. Specifically, SPRITE and ChIA-Drop do not fall into the category of "inferring multi-way interactions from pairwise interactions". They are experimental assays that can also provide information of multi-way interactions directly in individual nuclei.

2. The authors also fundamentally misunderstood the method MATCHA (Zhang and Ma 2020).

- In the paper, they wrote: "Prior work on neural networks highlights the utility of hypergraph representation learning to denoise and analyze multi-way contacts inferred from pairwise contact data and to predict de novo multi-way contacts [14]." This is false. MATCHA also utilizes ChIA-Drop and SPRITE as input to denoise the data and enhance the analysis of multi-way chromatin interactions.

- In the response to Reviewer #1, they wrote "In terms of predicting de novo multi-way contacts, the authors state that MATCHA is only applied to ChIA-Drop data." This is also a mischaracterization. The MATCHA paper clearly showed that the hypergraph based method is applied to both SPRITE and ChIA-Drop data to enhance the multi-way chromatin interaction analysis and can be naturally extended to Pore-C. The method in this manuscript is also relying on input multi-way assay data (Pore-C). Therefore, a careful comparison of the methods is essential.

- The authors need to study the existing literature carefully.

3. In the paper, the authors still do not provide detailed comparisons on both available data (SPRITE data on the same cell line) and method (MATCHA and other approaches) as I previously requested. There was only a brief (and inaccurate) description of these previous methods in the paper. Therefore, the methodological advances in practice of this work are entirely unclear and the quality of the identified multi-way interactions from Pore-C is also unclear.

4. The authors did not provide additional materials to demonstrate the novelty and rigor of the proposed computational workflow. Again, the way to identify multi-way contact in this work is a

straightforward counting/listing using the incidence matrix. The approaches do not have computational advances compared to prior work. The authors' approaches of comparing hypergraphs are also technically wrong and overall lack rigor. I further elaborate my points below on the technical details.

5. The authors adapted multiple approaches from existing methods already used for SPRITE and ChIA-Drop data to Pore-C without clearly showing why such adaptation is non-trivial.

- The authors argued that the construction of hyperedges from Pore-C is a major strength of this manuscript but the exact same idea of using hypergraph to represent multi-way chromatin interaction has already been used in MATCHA (which the authors mischaracterized in this work; mentioned above).

- The authors also stated that "the MATCHA algorithm has not been validated on Pore-C data" and "It is outside of the scope of our manuscript to adapt a computational algorithm to fit a new data type". However, both reviewers pointed out that Pore-C as a multi-way chromatin interaction data is closely related to the existing multi-way chromatin interaction data (e.g., SPRITE, ChIA-Drop). The authors should systematically demonstrate why Pore-C data differs from these existing multi-way data, and more importantly, carefully show that this difference makes the construction of hyperedges non-trivial as compared to existing computational approaches. These questions were not addressed.

6. The authors stated that using incidence matrix-based representation instead of adjacency matrix representation is a technical novelty. However, the so-called incidence matrix is essentially the same as "clusters" in SPRITE or "complexes" in ChIA-Drop. While not using the exact same term "incidence matrix", both methods used in the original publications use the same or very similar approach to visualize and analyze the data. For instance, see Fig. 2/3 in the ChIA-Drop paper: Zheng et al. Nature 2019. Thus the advance of incidence matrix analysis approach is unclear.

7. The so-called hypergraph entropy is calculated from $H H^T$ size of $n \times n$, where H is the incidence matrix and n is the number of nodes. By this design, this Laplacian matrix of this hypergraph is exactly the same as the adjacency matrix from the decomposed hypergraph, which makes it no difference from calculating the entropy for a pairwise chromatin interaction data such as Hi-C or decomposed Pore-C.

I took a closer look at the method that the authors cited (Bloch et al.). The original hypergraph entropy is calculated from $H^T H$ size of $m \times m$, where m is the number of hyperedges. Therefore, the definition and conclusion of this part of the analysis are technically wrong and the results are not reliable.

8. The authors' definition of hypergraph distance is flawed. There is no mathematical derivation or proof that such measurement can reflect the similarity between two graphs. Here I provide counterexamples. For the simplest case, consider G_1 contains only one hyperedge connecting node 1/2/3, G_2 contains only one hyperedge connecting node 4/5/6, the results eigenvalues for both hypergraphs would be $[1,1,1,1,1,0]$. Thus, the distance would be 0, indicating these two graphs are exactly the same, which is clearly incorrect. This metric would only make some sense if it measures the isomorphic similarity or permutation invariant similarity (cases where the id for nodes in two graphs do not directly correspond to each other but are known to be the same set). But I cannot see how that can be applied to the 3D genome graph as different nodes or different genomic loci directly correspond to each other based on genome coordinates and are not interchangeable with each other. Therefore, the hypergraph distance metric is flawed.

9. The authors did not answer my questions listed in points #3 and #4 in the previous review. The authors were talking about something else in their response, transcription clusters defined by their approach, while I was asking the reliability of these findings compared to previous literature and their biological significance. But given the flawed and problematic methods in this work, the biological findings are unlikely reliable. However, I list my questions again:

- What do these identified multiway contacts imply from Pore-C as compared to other methods?
- How do we know they are solid and reliable? (based on the problems I identified above, they aren't).

- What type of noise is there in Pore-C compared to other methods?

- What's the connection to other 3D genome structural and functional properties? The authors added new materials on analyzing the structural and functional properties of transcription clusters in terms of their involvement of CTCF and cohesin as well as a few cases of TFs. These analyses again simply listed the number of clusters and the names of TFs based on motif analysis. The figures with examples are not very informative. It is unclear what the statistical and functional significance of the results are. Also, the authors did not consider my suggestion to bring in orthogonal structural and functional datasets to support the findings to enhance the rigor.
- What's the difference compared to grouping pairwise contacts inferred from Hi-C? For example, this is conceptually similar to the method MOCHI (Tian et al. Genome Res 2020), which was mentioned in the previous review but the authors did not consider relevant, and a more recent study Yi et al. iScience 2021 (PMID: 34888502).

Response to Reviewer Comments on Manuscript NCOMMS-21-35046A-Z

Title: Deciphering Multi-way Interactions in the Human Genome

We thank the editor and the Reviewers for their time and valuable comments, and here we address the issues raised by the reviewers. Please see the following reviewer comments (in black) and our point-by-point responses (in blue). Based on the reviewers' suggestions, we have revised portions of our manuscript and addressed errors in the text.

Comments by Reviewer 1

I'm satisfied with the authors' responses.

Thank you very much for your feedback and prior guidance.

Comments by Reviewer 2

The authors did not make an effort to carefully address the comments from this reviewer. The main problems remain: 1) The analytic approaches used in this work lack novelty and rigor; and some methods are technically flawed, e.g., hypergraph entropy and hypergraph distance; 2) There is a major lack of detailed comparisons with prior work and data; 3) The biological analysis is inadequate as a research paper. Based on the responses to review comments and the revised manuscript, the authors appear to have incorrect and insufficient understanding of the existing work in chromatin biology and computational techniques. Overall, this work as it currently stands does not have sufficient conceptual advances in the field of 3D chromatin organization. In particular, the computational methods lack technical novelty and rigor and the manuscript still has not demonstrated that the methods have theoretical and practical evidence to produce reliable results. My main comments are listed below.

1. It is disappointing that the authors have not provided sufficient comparisons with existing work and available data as I previously pointed out. The authors' responses to the review also incorrectly represented prior work. The authors mentioned that "Recent methods for analyzing Hi-C data have inferred multi-way interactions from pairwise interactions[4-9], though with the limitation of some ambiguity inherent to artificial assembly of pairwise ligation junctions or machine learning-based predictions." and cited not only the computational methods that infer multi-way interactions from pairwise interactions (Olivares-Chauvet et al.[4] and Liu et al. [8]) but also include new experimental assays that are ligation free and capture multiway interactions directly (such as SPRITE: Quinodoz et al., ChIA-Drop: Zheng et al., a review paper: Kempfer et al., protocol for SPRITE: Quinodoz et al.). This clearly is a misrepresentation of the existing work in the literature. Specifically, SPRITE and ChIA-Drop do not fall into the category of "inferring multi-way interactions from pairwise interactions". They are experimental assays that can also provide information of multi-way interactions directly in individual nuclei.

Thank you for pointing out this oversight on our part. We revised our acknowledgement and description of these cited works beginning on line 19.

2. The authors also fundamentally misunderstood the method MATCHA (Zhang and Ma 2020).
 - In the paper, they wrote: "Prior work on neural networks highlights the utility of hypergraph representation learning to denoise and analyze multi-way contacts inferred from pairwise contact data and to predict de novo multi-way contacts [14]." This is false. MATCHA also utilizes ChIA-Drop and SPRITE as input to denoise the data and enhance the analysis of multi-way chromatin interactions.

We have changed our wording from "multi-way contacts inferred from pairwise contact data" to "existing multi-way contact data" on line 32 of the manuscript. Because SPRITE and ChIA-Drop both rely on paired-end sequencing, the sequencing reads generated by these assays do indeed capture pairwise contacts that are then grouped together with other paired contacts sharing the same distinct barcode. In comparison, Pore-C reads contain more than a pair of ligation junctions and thus directly capture multi-way contacts without requiring an assembly step like in SPRITE and ChIA-Drop. Therefore, we initially interpreted SPRITE clusters and ChIA-Drop complexes as inferred multi-way contacts.

- In the response to Reviewer 1, they wrote "In terms of predicting de novo multi-way contacts, the authors state that MATCHA is only applied to ChIA-Drop data." This is also a mischaracterization. The MATCHA paper clearly showed that the hypergraph based method is applied to both SPRITE and ChIA-Drop data to enhance the multi-way chromatin interaction analysis and can be naturally extended to Pore-C. The method in this manuscript is also relying on input multi-way assay data (Pore-C). Therefore, a careful comparison of the methods is essential.

Ulahannan et al. [1], who we cite extensively in our manuscript, developed the Pore-C protocol that we use and generated a multi-way contact dataset for the same GM12878 cell line. In their paper, they provide a

comprehensive comparison between the multi-way contacts captured by Pore-C and those extracted from the SPRITE pipeline (see Figure 2 from their paper). They concluded that: “SPRITE clusters were relatively depleted in multi-way contacts, and comprised a higher fraction of pairwise contacts (13.8%) and singletons (74.5%) compared to Pore-C (15.6% and 6.3%, respectively)”, that there was an “increased compartment specificity among Pore-C HOLR [high-order and long-range] contacts relative to SPRITE”, and that “the combinatorial distance decay we have shown [for] higher-order Pore-C contacts is sharper than what has been previously shown for SPRITE or ChIA-drop”. Given this extensive comparison, we do not believe that it is our responsibility to reproduce what has already been shown in this work, especially when the aim of our present study is to evaluate the transcriptional properties of the naturally-occurring multi-way chromatin interactions that we directly capture from the nucleus with long-read sequencing. As for applying MATCHA to our data, we do not require a denoising method as we have our own approach for filtering noise (see the ‘Hypergraph Filtering’ section in our Methods). Further, in the two years since its publication, MATCHA has not been applied to any other group’s published data – as evidenced by its current citations – suggesting that their denoising algorithm is not the gold-standard that the reviewer promotes it to be. For good measure, however, we did attempt to run MATCHA on our data and found the code and documentation to be incomprehensible. We reached out to the authors and asked if they could run their pipeline on a sample dataset for us or offer further guidance for running their software - they fulfilled neither request.

- The authors need to study the existing literature carefully.

Thank you for this suggestion.

3. In the paper, the authors still do not provide detailed comparisons on both available data (SPRITE data on the same cell line) and method (MATCHA and other approaches) as I previously requested. There was only a brief (and inaccurate) description of these previous methods in the paper. Therefore, the methodological advances in practice of this work are entirely unclear and the quality of the identified multi-way interactions from Pore-C is also unclear.

Thank you for your comment. Please see our response to Comment 2 (point 2) above for an explanation as to why we do not include a comparison to SPRITE or test MATCHA in our manuscript. As for the methodological advances of our work, we present a new way of classifying biologically relevant multi-way contacts, which we term transcription clusters, by combining our Pore-C data with other available -omics data. We also implement robust measures like hypergraph entropy and multiple hypergraph distance measures to quantify local and global complex chromatin structure across cell types.

4. The authors did not provide additional materials to demonstrate the novelty and rigor of the proposed computational workflow. Again, the way to identify multi-way contact in this work is a straightforward counting/listing using the incidence matrix. The approaches do not have computational advances compared to prior work. The authors’ approaches of comparing hypergraphs are also technically wrong and overall lack rigor. I further elaborate my points below on the technical details.

We do not claim to create a new computational framework in this study. Nor do we present a method for identifying multi-way contacts – multi-way contacts in this study are given, as they are the direct output of the Pore-C protocol [1] that we implement. What we do propose is the novel application of established and rigorous mathematical principles in analyzing multi-way chromatin interaction data. Additionally, we do in fact contribute computational advances compared to prior work. While MATCHA demonstrates that they can successfully denoise multi-way contact data and improve overall data quality by showing contact map comparisons between original SPRITE, denoised SPRITE, and Hi-C, as well as MIA-Sig-denoised ChIA-Drop, MATCHA-denoised ChIA-Drop, and original ChIA drop, they only show this intra-chromosomally. They state that their method makes long-range chromatin interaction features clearer through this denoising but fail to show in the figures or text how their method holds up inter-chromosomally. In our manuscript, we present clear examples of both intra-chromosomal and inter-chromosomal regions of multi-way contacts genome-wide. Additionally, MATCHA applies their method to 1 Mb resolution SPRITE data and 5 kb resolution ChIA-Drop data and later compares their denoised decomposed pairwise contact matrices at 100 kb resolution for SPRITE and 20 kb resolution for ChIA-Drop. While these resolutions are strong, we offer additional analyses at the read-level. With regard to the validity and rigor of our approach for comparing hypergraphs, please see response to comment 8 below.

5. The authors adapted multiple approaches from existing methods already used for SPRITE and ChIA-Drop data to Pore-C without clearly showing why such adaptation is non-trivial.
 - The authors argued that the construction of hyperedges from Pore-C is a major strength of this manuscript but the exact same idea of using hypergraph to represent multi-way chromatin interaction has already been used in MATCHA (which the authors mischaracterized in this work; mentioned above).

We acknowledge in the introduction section of our paper that MATCHA previously used hypergraph representation to denoise and analyze multi-way contacts. We distinguish our use of hypergraph representation

by including in our framework, the computation of hypergraph entropy and multiple hypergraph distance measures, both of which have not been applied in a biological context before outside of our work. These are robust metrics that offer unique perspectives on chromatin complexity and genome organization.

- The authors also stated that “the MATCHA algorithm has not been validated on Pore-C data” and “It is outside of the scope of our manuscript to adapt a computational algorithm to fit a new data type”. However, both reviewers pointed out that Pore-C as a multi-way chromatin interaction data is closely related to the existing multi-way chromatin interaction data (e.g., SPRITE, ChIA-Drop). The authors should systematically demonstrate why Pore-C data differs from these existing multi-way data, and more importantly, carefully show that this difference makes the construction of hyperedges non-trivial as compared to existing computational approaches. These questions were not addressed.

We are unable to compare our identified multi-way contacts against those presented in ChIA-Drop due to the species difference - they characterize multi-way contacts in *Drosophila*. They have since adapted their protocol to the GM12878 cell line but that data is currently unpublished. We note that the results of that study, once published, would be an appropriate comparison to our data. As mentioned in our response to comment 2, a Pore-C and SPRITE comparison has already been performed [1] and it was demonstrated that the Pore-C method of capturing multi-way contacts outperforms that of SPRITE. Additionally, as mentioned in our response to comment 2, we have attempted to run MATCHA on our Pore-C data without success. We further anticipate that the MATCHA software is currently incompatible with our read-level data.

6. The authors stated that using incidence matrix-based representation instead of adjacency matrix representation is a technical novelty. However, the so-called incidence matrix is essentially the same as “clusters” in SPRITE or “complexes” in ChIA-Drop. While not using the exact same term “incidence matrix”, both methods used in the original publications use the same or very similar approach to visualize and analyze the data. For instance, see Fig. 2/3 in the ChIA-Drop paper: Zheng et al. Nature 2019. Thus the advance of incidence matrix analysis approach is unclear.

An incidence matrix is a mathematical representation of data showing the relationship between two classes of objects - vertices and edges [2]. This is in contrast to an adjacency matrix which represents the relationship between pairs of vertices. We distinctively preserve the structure of the data and perform our computations on an incidence matrix, whereas in ChIA-Drop, what appears to be a similar representation used in their figures is merely for multi-fragment visualization, as indicated in their companion paper [3]. While an incidence matrix is a data structure that could hold SPRITE clusters and ChIA-Drop complexes or any other notion of a multi-way interaction, it is critical to note that it is two very different procedures to compute a distance measure, for example, from an incidence matrix than from the decomposed adjacency matrices that SPRITE and ChIA-Drop construct for their analyses.

7. The so-called hypergraph entropy is calculated from HH^T size of $n \times n$, where H is the incidence matrix and n is the number of nodes. By this design, this Laplacian matrix of this hypergraph is exactly the same as the adjacency matrix from the decomposed hypergraph, which makes it no difference from calculating the entropy for a pairwise chromatin interaction data such as Hi-C or decomposed Pore-C. I took a closer look at the method that the authors cited (Bloch et al.). The original hypergraph entropy is calculated from $H^T H$ size of $m \times m$, where m is the number of hyperedges. Therefore, the definition and conclusion of this part of the analysis are technically wrong and the results are not reliable.

Thank you for pointing out this error. We have changed the definition of the Laplacian matrix in computing hypergraph entropy. The new definition is consistent with the normalized Laplacian matrix defined in hypergraph distance, which has been used in the most hypergraph learning works [4, 5]. We observe that the fibroblast hypergraphs have higher entropy than the B lymphocyte hypergraph which is consistent with the potency of these two lineages, as fibroblasts are less specialized than B lymphocytes and thus incur more potential and disorder. Additionally, adult and neonatal fibroblasts have more similar entropy values across chromosomes as compared to B lymphocytes, which we expect given that they are the same cell type just with different maturity.

8. The authors' definition of hypergraph distance is flawed. There is no mathematical derivation or proof that such measurement can reflect the similarity between two graphs. Here I provide counterexamples. For the simplest case, consider G_1 contains only one hyperedge connecting node 1/2/3, G_2 contains only one hyperedge connecting node 4/5/6, the results eigenvalues for both hypergraphs would be $[1, 1, 1, 1, 1, 0]$. Thus, the distance would be 0, indicating these two graphs are exactly the same, which is clearly incorrect. This metric would only make some sense if it measures the isomorphic similarity or permutation invariant similarity (cases where the id for nodes in two graphs do not directly correspond to each other but are known to be the same set). But I cannot see how that can be applied to the 3D genome graph as different nodes or different genomic loci directly correspond to each other based on genome coordinates and are not interchangeable with each other. Therefore, the hypergraph distance metric is flawed.

Thank you for your input. Spectral distance, which involves comparing graphs based on eigenvalues of the graph Laplacian, is a well established approach in the literature, see [6] and references therein. Here we are leveraging that approach for hypergraph comparison by converting hypergraph into a graph representation, see [7] for details. The reviewer is correct in pointing out that spectral distance is not node permutation invariant and thus does not preserve node identities while comparing two graphs. The counter example which the reviewer gave, in fact involves two isomorphic graphs, i.e, one graph is the same as the other by simply renumbering its node number. However, the distribution of eigenvalues of the graph Laplacian is related to global structures in the graph, such as number of spanning trees, community structure, etc. [6], and thus captures global differences in the graph. Thus, while spectral distance ignores node identities, it can still be useful to capture differences at global scale in the graph. To this regard, in the revised manuscript, we have also included two additional distance measures: Hamming distance and DeltaCon distance (see 'Hypergraph Distance' section). Hamming distance preserves node identities and captures local differences in neighbourhood of the nodes. On the other hand, DeltaCon also captures global connectivity structure in terms of how information flows between any two nodes (which is related to global connectivity of the graph), and thus captures both local and global differences between the graphs, while preserving node identities. We have also added a discussion to highlight such differences between the distance measures in the Supplemental Notes section.

9. The authors did not answer my questions listed in points 3 and 4 in the previous review. The authors were talking about something else in their response, transcription clusters defined by their approach, while I was asking the reliability of these findings compared to previous literature and their biological significance. But given the flawed and problematic methods in this work, the biological findings are unlikely reliable. However, I list my questions again:

- What do these identified multiway contacts imply from Pore-C as compared to other methods?

Our multiway-contacts and subsequent classification into transcription clusters is conceptually similar to SPRITE clusters, ChIA-Drop complexes, and the notion of transcription factories or hubs - where each reflect the behavior of genomic loci flocking together to access the same transcriptional machinery and coordinate expression. Uniquely, we were able to provide an exhaustive atlas of intra- and inter-chromosomal multi-way contacts along with their properties across three cell types, extracting insights into cell type-specific transcription clusters, TFs previously shown to be cell type-specific and that interact uniquely with transcription clusters in that cell type, architectural coupling, and transcription clusters with organization that suggest the ability to undergo self-sustaining regulation.

- How do we know they are solid and reliable? (based on the problems I identified above, they aren't).

Many 3C-based methods for mapping genome architecture leverage DNA-FISH to validate putative contacts as summarized in [8]. This strategy has previously been extended to Pore-C data as demonstrated in [1] when authors confirmed the presence and amplification of a cancer rearrangement junction captured as a Pore-C concatemer containing loci from chromosomes 9, 12, and 20 in the breast cancer cell line HCC1954 using DNA-FISH.

- What type of noise is there in Pore-C compared to other methods?

Spurious ligation products are the primary source of noise that we contend with in this data. As laid out in the 'Hypergraph Filtering' section of our Methods, we filter out (denoise) these false positive multiway contacts by decomposing the multiway contacts into their pairwise components and removing contacts that fall outside of the 85th percentile of contact frequency.

- What's the connection to other 3D genome structural and functional properties? The authors added new materials on analyzing the structural and functional properties of transcription clusters in terms of their involvement of CTCF and cohesin as well as a few cases of TFs. These analyses again simply listed the number of clusters and the names of TFs based on motif analysis. The figures with examples are not very informative. It is unclear what the statistical and functional significance of the results are. Also, the authors did not consider my suggestion to bring in orthogonal structural and functional datasets to support the findings to enhance the rigor.

In previous work from our group [9], we established the relationship between genome structure and function. Here, we go beyond that and define transcription clusters - inspired by the classical notion of coordinated transcription facilitated by multiple chromatin loci sharing space, dubbed 'transcription factories' by Peter Cook, [10]. We meticulously curated a TF-DNA interaction matrix that captures the relationship between transcription factors and genes based on TF binding motifs, chromatin accessibility, and regulatory influence (activating or inhibitory) [11] and integrated this information with long-read sequencing data to systematically extract modules of expression from complex, higher-order genome organization. As we state in our discussion, "Exploring long-range, inter-chromosomal interactions genome-wide offers the opportunity to establish fundamental principles of genome organization. Unbiased capture and study of multi-way contacts can help identify biologically important assemblies that affect transcription, such as transcription clusters This approach

can also connect genome organization principles to the study of transcription factors and how they govern cell type-specific network architecture.” Moreover, the implications of “simply listing the number of clusters and names of TFs”, among the quantitative analyses we provide, are a deeper understanding of the relationship between transcription factors and genome structure which is integral to advancements in characterizing genome dynamics and controllability [12].

We already state the functional significance of preferential CTCF and cohesin binding at transcription clusters in the manuscript, “these data suggest that the identified transcription clusters are important sites of transcriptional regulation, and support a model in which CTCF and cohesin actively mediate multi-way interactions”. As for statistical significance, we now include p-values with these findings. We further highlight that uniquely-binding TFs that we identified may be, “involved in cell type-specific regulation in transcription clusters” and point out that a number of them have previously been described to have cell-type specific roles in prior studies.

- What’s the difference compared to grouping pairwise contacts inferred from Hi-C? For example, this is conceptually similar to the method MOCHI (Tian et al. *Genome Res* 2020), which was mentioned in the previous review but the authors did not consider relevant, and a more recent study Yi et al. *iScience* 2021 (PMID: 34888502).

Thank you for highlighting these additional methods. We have already emphasized in our manuscript and the previous round of reviewer responses, that there is a clear advantage in directly capturing multi-way contacts than inferring them from pairwise data. It is inevitable that grouped pairwise contacts will over- or underestimate the actual number of naturally-occurring multi-way contacts more severely than assays like Pore-C that enable direct capture of multiple simultaneously occurring chromatin contacts.

References

- [1] Netha Ulahannan, Matthew Pendleton, Aditya Deshpande, Stefan Schwenk, Julie M Behr, Xiaoguang Dai, Carly Tyer, Priyesh Rughani, Sarah Kudman, Emily Adney, et al. Nanopore sequencing of dna concatemers reveals higher-order features of chromatin structure. *bioRxiv*, page 833590, 2019.
- [2] Daniel Zelazo, Mehran Mesbahi, and Mohamed-Ali Belabbas. Graph theory in systems and controls. In *2018 IEEE Conference on Decision and Control (CDC)*, pages 6168–6179. IEEE, 2018.
- [3] Simon Zhongyuan Tian, Daniel Capurso, Minji Kim, Byoungkoo Lee, Meizhen Zheng, and Yijun Ruan. Chia-dropbox: a novel analysis and visualization pipeline for multiplex chromatin interactions. *bioRxiv*, page 613034, 2019.
- [4] Dengyong Zhou, Jiayuan Huang, and Bernhard Schölkopf. Learning with hypergraphs: Clustering, classification, and embedding. *Advances in neural information processing systems*, 19, 2006.
- [5] Yue Gao, Zizhao Zhang, Haojie Lin, Xibin Zhao, Shaoyi Du, and Changqing Zou. Hypergraph learning: Methods and practices. *IEEE Transactions on Pattern Analysis and Machine Intelligence*, 2020.
- [6] Claire Donnat and Susan Holmes. Tracking network dynamics: A survey using graph distances. *The Annals of Applied Statistics*, 12(2):971 – 1012, 2018.
- [7] Amit Surana, Can Chen, and Indika Rajapakse. Hypergraph similarity measures. *arXiv preprint arXiv:2106.08206*, 2021.
- [8] Rieke Kempfer and Ana Pombo. Methods for mapping 3d chromosome architecture. *Nature Reviews Genetics*, 21(4):207–226, 2020.
- [9] Haiming Chen, Jie Chen, Lindsey A Muir, Scott Ronquist, Walter Meixner, Mats Ljungman, Thomas Ried, Stephen Smale, and Indika Rajapakse. Functional organization of the human 4d nucleome. *Proceedings of the National Academy of Sciences*, 112(26):8002–8007, 2015.
- [10] David RF Carter, Christopher Eskiw, and Peter R Cook. Transcription factories. *Biochemical Society Transactions*, 36(4):585–589, 2008.
- [11] Scott Ronquist, Geoff Patterson, Lindsey A Muir, Stephen Lindsly, Haiming Chen, Markus Brown, Max S Wicha, Anthony Bloch, Roger Brockett, and Indika Rajapakse. Algorithm for cellular reprogramming. *Proceedings of the National Academy of Sciences*, 114(45):11832–11837, 2017.
- [12] Indika Rajapakse and Steve Smale. Mathematics of the genome. *Foundations of Computational Mathematics*, 17(5):1195–1217, 2017.

Reviewers' Comments:

Reviewer #2:

Remarks to the Author:

The authors made minimal effort to address the major concerns and flaws in this work. I think that this work as it currently stands, several major issues remain:

1. There is a lack of appropriate and rigorous mathematical methods in this work.
2. There is a lack of understanding of the existing methods in the field, leading to incorrect conclusions of the analysis advances.
3. Overall, there is a lack of innovation in this work to demonstrate advances, in both computational methods and insights from biological data analysis, as previously discussed.

The authors argued that they do propose "novel application of established and rigorous mathematical principles in analyzing multi-way chromatin interaction data." I do not agree. As I pointed before, there are serious issues with the math in this work, and the authors did not address them. Even though the mathematical methods employed by the authors have long been established, the ways that the authors used these math techniques are not correct. Just because the method and the problem both have the word "hypergraph", it does not mean that the method being employed is appropriate to use for the specific type of data in this work.

1. The hypergraph entropy was proposed and defined in Block et al. with the H^{TH} . All properties of hypergraph entropy were proved in Block et al. conditioned on that it is calculated with H^{TH} . However, the authors used HH^T instead. Even though both of them are referred to as the Laplacian matrix, they are totally different. The cases that the authors cited where HH^T was used instead, they did not involve the calculation of hypergraph entropy, and thus not relevant here. In other words, the authors did not use the hypergraph entropy correctly in this work. I pointed this out previously but the authors did not address.

2. Importantly, one major problem of using HH^T is that it is in fact entirely equivalent to decomposing each hyperedge into edges, resulting in the adjacency matrix, which is exactly what SPRITE and ChIA-Drop did in their published work. The conclusions derived from this type of metric are thus expected and not related to hypergraphs whatsoever.

3. For the spectral distance, it is again a well established approach, on the condition that the node identity is not relevant, which is not the case here as I pointed out previously. The authors did not address. The two newly proposed distances are based on the adjacency matrix only. Therefore, they are not related to hypergraph.

Overall, there are fundamental flaws in the math formulation in this work. The application of hypergraph based methods is not correct as the authors claimed. On the other hand, the approaches that are reasonable are equivalent to studying pairwise interactions, thereby not showing the advantages of multi-way chromatin interactions provided by Pore-C.

In terms of the advances in this work, I think it's too thin and too scattered. The authors need to have a complete change to refocus the work and analysis. The authors should not give a new name to existing work in the field.

1. Representing the multi-way chromatin interaction as hypergraph has been done before. The representation of the incidence matrix is equivalent to what SPRITE and ChIA-Drop did. The authors said that what ChIA-Drop did is just a multi-fragment visualization. The same argument can indeed also apply to the authors' "read-resolution" hypergraphs -- it's just a visualization.

2. Applying established mathematical methods on hypergraph for the hypergraph constructed from the multi-way data need to perform in the correct way. As discussed before, many of the critical hypergraph mathematical methods described by the authors were all used in an incorrect way in this work. For those that are somewhat appropriate, they are based on the decomposed adjacency matrix.

From the authors' responses, it seems that the authors incorrectly thought that the following two

approaches are different:

- a) In ChIA-Drop and SPRITE, the authors originally decomposed interactions first, represented the data as an adjacency matrix, and analyzed the matrix.
- b) In this work, the authors first represent the data as an incidence matrix, then use HH^T to transform that into an adjacency matrix, then apply methods on that. They claim that because the calculation of HH^T and the following procedure is done on H , which is the incidence matrix. However, HH^T is just the same as decomposing interactions.

Title: Deciphering Multi-way Interactions in the Human Genome

We thank the editor, the editorial team, and the reviewers for their time and input on this manuscript. Here we emphasize our point-by-point responses to reviewer comments from the most recent round of review. Reviewer comments (black text) are addressed point-by-point (**bold blue text**) below.

Point-by-point for NCOMMS-21-35046B-Z — April 2022

Comments by Reviewer 2

The authors made minimal effort to address the major concerns and flaws in this work. I think that this work as it currently stands, several major issues remain:

1. There is a lack of appropriate and rigorous mathematical methods in this work.

We appreciate the reviewer's candor and hope to demonstrate here the suitability of our methods.

(a) Regarding the first claim, of lack of appropriate mathematical methods, there is ambiguity in this statement, so we interpret the reviewer's meaning to be that they think the methods in the present work are not appropriately used rather than that the present work is *missing* a methodological approach. We first note that our definitions for hypergraphs and higher-order chromatin contacts are consistent with the literature, e.g., (1). Methods for understanding higher-order chromatin structure from data are not well-established in the field, and use of hypergraphs to help us understand the data is very promising (1) and should be explored for its utility in different datasets, including Pore-C which has not been done. In our use of hypergraph theory with Pore-C data, we point to its appropriate application in that the outcomes are consistent with the known biology of the systems. One validation we have used is comparing TAD structures identified based on our incidence matrices to TADs that we previously identified from Hi-C data from the same cell type, which had consistently high overlap. We note that the hypergraph representations used are useful for visualization as well as computation, as they preserve higher-order structure, a point that is expanded on below. Our approach made use of this preservation of data structure to do additional computations, such as distance and entropy between datasets. Consistent with the biology, hypergraph distance captured broad lineage relationships among different cell types, showing that the multi-way contacts in neonatal and adult fibroblasts were much more similar to each other than to the multi-way contacts in B cells. These findings suggest an appropriate method that returns biologically meaningful results.

(b) Regarding the claim of 'lack of mathematical rigor', we feel that the mistake we made in defining hypergraph entropy in our original submission led the reviewer to lose credibility in our work. We corrected the definition of the Laplacian for hypergraph entropy from $L = HH^T$ to be $L = D - HE^{-1}H^T$, where $D \in \mathbb{R}^{n \times n}$ is a diagonal matrix containing the degrees of nodes along its diagonal, and $E \in \mathbb{R}^{m \times m}$ is a diagonal matrix containing the orders of hyperedges along its diagonal as in (2–11). This hypergraph Laplacian is driven from the real-valued relaxation to approximately obtain hypergraph normalized cuts, similar to the definition of Laplacian matrices for graphs (5). Although the two definitions look similar, the two degree matrices D and E contain higher-order structural information of the hypergraph. Computations and biological interpretations of the results using this definition have been carefully checked based on reviewer feedback.

2. There is a lack of understanding of the existing methods in the field, leading to incorrect conclusions of the analysis advances.

We thank the reviewer for their feedback. To ensure clarity in our introduction we have revised the writing in the Introduction (beginning on line 19) to more clearly acknowledge previous multi-way chromatin contact studies. We summarize the major related alternatives to the Pore-C assay below. To our knowledge, only one closely-related study (1) deals explicitly with hypergraph representations of higher-order chromatin structure and does so trivially, without exploiting well-accepted mathematical theory to elucidate the properties of the hypergraph. We claim that the use of hypergraph representation of Pore-C assays is novel, and is a natural interpretation of Pore-C concatemers. We are aware of no other studies that explicitly represent Pore-C concatemers using hypergraph theory. We provide a more exhaustive summary of drawbacks in comparison to Pore-C and our analytical framework below.

- **Tethered multiple 3C (TMC3) (12) uses a two-phase mapping approach to sequentially identify paired ends of ligated chimeric sequences, first mapping the original ends and then cleaving the read and identifying the newly resulting ends. The number of uniquely mapped regions in a read are then retroactively classified as pairwise or multi-way contacts. TMC3 characterized and analyzed these contacts for two cell lines, extracting features of hierarchical genome organization (contact decay with genomic distance, chromosomal compartments, topological domains) commonly described for**

pairwise contact data. Size selection of 250bp and short read sequencing limits the identification of TMC3 multi-way contacts to an order of 4, however.

- C-walks (13) cleverly infer multi-way contacts from 3C data by computationally assembling chains of ligation junctions from shotgun sequencing. They also introduce a sub-method, 3way-4C, inferring 3-way C-walks from 4C data to look more closely at 3-way interactions at select highly transcribed locations in active TADs. Multi-way inter-chromosomal interactions from their protocol are described as spurious and they primarily characterize pairwise chromosomal interactions and multi-way intra-chromosomal interactions. Most of their multi-way analysis is focused on local structure like inter-TAD interactions and 3-way C-walks at select genomic regions.
- COLA (14) takes advantage of an anomaly in the Hi-C procedure where the ligation of three or more nearby fragments in then nucleus are occasionally captured. COLA modifies the Hi-C protocol, digesting chromatin into finer fragments that they argue yield a higher proportion of reads containing three or more contacting loci. Though useful for extracting more from existing Hi-C datasets, the output is non-exhaustive of the existing multi-way contacts reported by several studies including our own, primarily finding high-quality 3-way contacts, up to a handful of 5-way contacts.
- Similar to Pore-C, MC-4C (15) uses Oxford Nanopore long sequencing reads to preserve multi-way contacts. Their protocol is designed for targeted analysis, capturing multi-way interactions for regions of interest instead of genome-wide. Similarly focused on local coverage, Tri-C (16) captures primarily pairwise and 3-way interactions from 3C reads at specific genomic regions of interest to interrogate regulatory hubs and conformational properties at participating promoters and enhancers.
- Liu et al.(17) use principles of Gaussian polymer chains and polymer-based structural modeling to infer multi-way contacts from Hi-C data. Authors benchmarked their inferred contacts against multi-way contacts extracted from methods mentioned above, reporting "good" correlation. While useful if Hi-C data is already available, inference of multi-way interactions for pairwise data remains an ill-posed problem regardless of the informed assumptions underlying their model.
- SPRITE (18) crosslinks DNA, splits the DNA into uniquely barcoded wells where covalently linked loci will sort together, then sequentially performs this split-pool and tag sequence again until complexes of interacting molecules have a distinct barcode. Tagged DNA is sequenced and reads with matching barcodes are clustered together as multi-way contacts. Similarly, ChIA-Drop (19) crosslinks DNA, fragments the DNA, and loads it into a microfluidic device where linked DNA are partitioned into the same droplet and tagged with a unique barcode. Once sequenced, reads with the same barcode are grouped together as multi-way contacts. Both SPRITE and ChIA-Drop are formidable approaches and have been previously benchmarked against Pore-C (20) where it was shown that, SPRITE clusters were relatively depleted in multi-way contacts and contained a higher proportion of pairwise contacts compared to Pore-C, that there was an increased compartment specificity among Pore-C high-order and long-range contacts relative to SPRITE, and that the combinatorial distance decay shown for higher-order Pore-C contacts was sharper than it was for SPRITE or ChIA-drop.

We contend that the present work advances the utility of the recently published Pore-C assay for meaningful higher-order analysis of chromatin structure (20). The mathematical tools we extend provide novel insights and serve as a theoretical and computational platform for future Pore-C studies, and possibly for similarly structured data obtained using other assays. We believe that the precision of higher-order chromatin measurement afforded by Pore-C will be a valuable asset to the study of chromatin organization, and that these data are naturally represented by mathematical objects with well-established theoretical tools.

3. Overall, there is a lack of innovation in this work to demonstrate advances, in both computational methods and insights from biological data analysis, as previously discussed.

We respectfully disagree with the reviewer's assessment that our work lacks innovation on the computational and biological fronts. Although previous studies have used hypergraph representations, they have not applied comparisons between hypergraphs. In the context of chromatin organization this is an important task, one which benefits from new tools like those we extend in the present work. Computationally, this work is the first to apply hypergraph theory to long-read sequencing data in the study of genome structure. Moreover, our use of Pore-C long-read sequencing data presents an advantage over existing studies as multi-way contacts are captured directly without the need for amplification, inference, or assembly in any way. Our utilization of incidence matrices to show higher-order cis and *trans* contacts is conceptually similar to visualizations in other studies. However, a major contribution of the present work is the *explicit* connection between incidence matrices and the higher-order Laplacian. Biologically, this work is the first to extract cell type-specific transcription clusters genome-wide directly from multi-way contact data. We demonstrate higher-order genome organization for two cell types (primary adult fibroblasts and a neonatal fibroblast cell line) not previously characterized in this context, in addition to the well-studied GM12878 cell line. Through our hypergraph framework, we extracted genome-wide, cell-type

specific signatures from the atlas of multi-way interactions produced by Pore-C, which we believe can be informative for clinical research.

The authors argued that they do propose "novel application of established and rigorous mathematical principles in analyzing multi-way chromatin interaction data." I do not agree. As I pointed before, there are serious issues with the math in this work, and the authors did not address them. Even though the mathematical methods employed by the authors have long been established, the ways that the authors used these math techniques are not correct. Just because the method and the problem both have the word "hypergraph", it does not mean that the method being employed is appropriate to use for the specific type of data in this work.

We certainly understand and apologize for not thoroughly explaining how the higher-order Laplacian fits into this context and our overall rationale for why we find that established hypergraph methods can be adapted in this work. The (normalized) hypergraph Laplacian used in this work (for hypergraph entropy and hypergraph distance) is a well established notion in the hypergraph learning community. It is driven from the real-valued relaxation to approximately obtain hypergraph normalized cuts, similar to the definition of Laplacian matrices for graphs (5). The eigenvalues of the hypergraph Laplacian can reveal the topological patterns (e.g., connectivity) of the hypergraph (5). The methods (i.e., hypergraph entropy and hypergraph distance) proposed in this work are developed based on the eigenvalues of the hypergraph Laplacian, both of which therefore can be used to quantify the structural properties of hypergraphs. The Pore-C data can be naturally represented by hypergraphs, where nodes are genomic loci and hyperedges are multi-way contacts among loci. Thus, our methods are appropriate for understanding the geometry of the human genome through the Pore-C data.

1. The hypergraph entropy was proposed and defined in Block et al. with the $H^T H$. All properties of hypergraph entropy were proved in Block et al. conditioned on that it is calculated with $H^T H$. However, the authors used HH^T instead. Even though both of them are referred to as the Laplacian matrix, they are totally different. The cases that the authors cited where HH^T was used instead, they did not involve the calculation of hypergraph entropy, and thus not relevant here. In other words, the authors did not use the hypergraph entropy correctly in this work. I pointed this out previously but the authors did not address.

Thank you for this response. We recall that you pointed this error out to us in the first round of revisions and appreciate the opportunity we were given to revise that error. Per your comment, we changed the definition of the Laplacian matrix in computing hypergraph entropy. The new definition is consistent with the normalized Laplacian matrix defined in hypergraph distance and used in most hypergraph learning works from hypergraph clustering to hypergraph neural networks (2–11). This hypergraph Laplacian shares many structural properties with the Laplacian matrices for graphs (e.g., the smallest eigenvalue of the hypergraph Laplacian is zero with the corresponding eigenvector filled with one) while keeping higher-order patterns of the hypergraphs (5). Therefore, using this hypergraph Laplacian has significantly improved the performance of hypergraph clustering and classification tasks in their works. Based on the literature and the formulation of graph entropy (i.e., computing the eigenvalue decomposition of the Laplacian matrix of a graph), our definition of hypergraph entropy is well-defined and can be used to detect structural properties of hypergraphs (e.g., the Pore-C data). Therefore, we computed the entropy of intra-chromosomal genomic hypergraphs for both fibroblasts and B lymphocytes at 1Mb scale. We find that the definition of hypergraph entropy we use is able to correctly quantify the higher-order structural features of the chromosomes for different cell types.

2. Importantly, one major problem of using HH^T is that it is in fact entirely equivalent to decomposing each hyperedge into edges, resulting in the adjacency matrix, which is exactly what SPRITE and ChIA-Drop did in their published work. The conclusions derived from this type of metric are thus expected and not related to hypergraphs whatsoever.

We thank the reviewer for their response. We currently use $HE^{-1}H^T$ and the resulting Laplacian ($L = D - HE^{-1}H^T$), not HH^T . Importantly, this definition is not equivalent to simply decomposing each hyperedge into its edge components, as it also considers the degrees of nodes and hyperedges (i.e., the two degree matrices D and E in the definition equation) of the hypergraph, which contain higher-order structural information of the hypergraph. Therefore, our resulting adjacency matrix preserves multi-way information, distinguishing our work from SPRITE and ChIA-Drop.

3. For the spectral distance, it is again a well established approach, on the condition that the node identity is not relevant, which is not the case here as I pointed out previously. The authors did not address. The two newly proposed distances are based on the adjacency matrix only. Therefore, they are not related to hypergraph.

We thank the reviewer for their response. We addressed the spectral distance concern by adding a remark in the revised manuscript that such a distance is not node permutation invariant. However, we would like to point out that when comparing graphs or hypergraphs, one can take different perspectives leading to different choices of similarity measures or distances. For instance, if one does not care about preserving node identities during comparison, the use of spectral distance is justified. On the other hand, if one

wants to preserve node identities during comparison, one could use adjacency matrix-based distances among many other choices. In fact we added two adjacency matrix-based distances and results based on that in the revised manuscript. Typically in applications, it is not uncommon to try different distances and see what each of them reveal. If the reviewer is not familiar with such a perspective, we would suggest that they look at the extensive body of literature on graph similarity measures. Specifically, we refer the reviewer to the paper (21) and references therein. To be clear, we do not agree with the reviewer's concern that the use of spectral distance is incorrect in our application – while spectral distance does not preserve node identities, it can still be useful in revealing global differences between two graphs such as those arising due to difference in number of disconnected components, hub/community structure, number of spanning trees, etc. (see (21) for detailed discussion).

Regarding the reviewer's second point: "The two newly proposed distances are based on the adjacency matrix only. Therefore, they are not related to hypergraph". As we pointed out above, that conversion of hypergraph into some form of a graph and using the associated adjacency matrix/Laplacian for hypergraph analysis is well-established in the literature (2–11). Therefore, we respectfully disagree with the reviewer's conclusion that adjacency matrix-based distances are not related or useful for hypergraph comparison. Our use of such distances is not only mathematically rigorous but also well-justified.

Overall, there are fundamental flaws in the math formulation in this work. The application of hypergraph based methods is not correct as the authors claimed. On the other hand, the approaches that are reasonable are equivalent to studying pairwise interactions, thereby not showing the advantages of multi-way chromatin interactions provided by Pore-C.

In terms of the advances in this work, I think it's too thin and too scattered. The authors need to have a complete change to refocus the work and analysis. The authors should not give a new name to existing work in the field.

1. Representing the multi-way chromatin interaction as hypergraph has been done before. The representation of the incidence matrix is equivalent to what SPRITE and ChIA-Drop did. The authors said that what ChIA-Drop did is just a multi-fragment visualization. The same argument can indeed also apply to the authors' "read-resolution" hypergraphs – it's just a visualization.

Simply put, our read-resolution hypergraphs are not limited to visualization. At the read-level, we do show hypergraphs in an incidence matrix for visualization but this matrix can also be used as input for hypergraph entropy and distance measures for further insight, as described above. It is important to note that our incidence matrix (how we mathematically represent hypergraphs) is both for visualization and computation whereas similar representations used in SPRITE and ChIA-Drop were only for visualization. We compute entropy and distance measures from an incidence matrix (capturing the relationship between vertices and hyperedges (22)) while SPRITE and ChIA-Drop perform computations from decomposed adjacency matrices (capturing the relationship between pairs of vertices) - the two provide different outcomes.

2. Applying established mathematical methods on hypergraph for the hypergraph constructed from the multi-way data need to perform in the correct way. As discussed before, many of the critical hypergraph mathematical methods described by the authors were all used in an incorrect way in this work. For those that are somewhat appropriate, they are based on the decomposed adjacency matrix.

From the authors' responses, it seems that the authors incorrectly thought that the following two approaches are different:

- In ChIA-Drop and SPRITE, the authors originally decomposed interactions first, represented the data as an adjacency matrix, and analyzed the matrix.
- In this work, the authors first represent the data as an incidence matrix, then use HH^T to transform that into an adjacency matrix, then apply methods on that. They claim that because the calculation of HH^T and the following procedure is done on H, which is the incidence matrix. However, HH^T is just the same as decomposing interactions.

To reiterate, we use $HE^{-1}H^T$ and the resulting Laplacian in our work ($L = D - HE^{-1}H^T$). This is not equivalent to simply decomposing each hyperedge into edges, as it also considers the degrees of nodes and hyperedges (i.e., the two degree matrices D and E in the Laplacian equation) of the hypergraph. This is an additional step that SPRITE and ChIA-Drop do not perform when constructing their adjacency matrix, thus our approaches are fundamentally different.

References

- [1] Ruochi Zhang and Jian Ma. Matcha: Probing multi-way chromatin interaction with hypergraph representation learning. *Cell systems*, 10(5):397–407, 2020.
- [2] Zizhao Zhang, Haojie Lin, Yue Gao, and KLISS BNRist. Dynamic hypergraph structure learning. In *IJCAI*, pages 3162–3169, 2018.
- [3] Song Bai, Feihu Zhang, and Philip HS Torr. Hypergraph convolution and hypergraph attention. *Pattern Recognition*, 110:107637, 2021.
- [4] Yifan Feng, Haoxuan You, Zizhao Zhang, Rongrong Ji, and Yue Gao. Hypergraph neural networks. In *Proceedings of the AAAI Conference on Artificial Intelligence*, volume 33, pages 3558–3565, 2019.
- [5] Dengyong Zhou, Jiayuan Huang, and Bernhard Schölkopf. Learning with hypergraphs: Clustering, classification, and embedding. *Advances in neural information processing systems*, 19, 2006.
- [6] Yue Gao, Zizhao Zhang, Haojie Lin, Xibin Zhao, Shaoyi Du, and Changqing Zou. Hypergraph learning: Methods and practices. *IEEE Transactions on Pattern Analysis and Machine Intelligence*, 2020.
- [7] Ze Tian, TaeHyun Hwang, and Rui Kuang. A hypergraph-based learning algorithm for classifying gene expression and arraycgh data with prior knowledge. *Bioinformatics*, 25(21):2831–2838, 2009.
- [8] Pei Dong, Yanrong Guo, Yue Gao, Peipeng Liang, Yonghong Shi, and Guorong Wu. Multi-atlas segmentation of anatomical brain structures using hierarchical hypergraph learning. *IEEE Transactions on Neural Networks and Learning Systems*, 31(8):3061–3072, 2019.
- [9] Xiao Zheng, Wenyang Zhu, Chang Tang, and Minhui Wang. Gene selection for microarray data classification via adaptive hypergraph embedded dictionary learning. *Gene*, 706:188–200, 2019.
- [10] Wei Shao, Yao Peng, Chen Zu, Mingliang Wang, Daoqiang Zhang, Alzheimer’s Disease Neuroimaging Initiative, et al. Hypergraph based multi-task feature selection for multimodal classification of alzheimer’s disease. *Computerized Medical Imaging and Graphics*, 80:101663, 2020.
- [11] Li Xiao, Junqi Wang, Peyman H Kassani, Yipu Zhang, Yuntong Bai, Julia M Stephen, Tony W Wilson, Vince D Calhoun, and Yu-Ping Wang. Multi-hypergraph learning-based brain functional connectivity analysis in fmri data. *IEEE transactions on medical imaging*, 39(5):1746–1758, 2019.
- [12] Ferhat Ay, Thanh H Vu, Michael J Zeitz, Nelle Varoquaux, Jan E Carette, Jean-Philippe Vert, Andrew R Hoffman, and William S Noble. Identifying multi-locus chromatin contacts in human cells using tethered multiple 3c. *BMC genomics*, 16(1):1–17, 2015.
- [13] Pedro Olivares-Chauvet, Zohar Mukamel, Aviezer Lifshitz, Omer Schwartzman, Noa Oded Elkayam, Yaniv Lubling, Gintaras Deikus, Robert P Sebra, and Amos Tanay. Capturing pairwise and multi-way chromosomal conformations using chromosomal walks. *Nature*, 540(7632):296–300, 2016.
- [14] Emily M Darrow, Miriam H Huntley, Olga Dudchenko, Elena K Stamenova, Neva C Durand, Zhuo Sun, Su-Chen Huang, Adrian L Sanborn, Ido Machol, Muhammad Shamim, et al. Deletion of dxz4 on the human inactive x chromosome alters higher-order genome architecture. *Proceedings of the National Academy of Sciences*, 113(31):E4504–E4512, 2016.
- [15] Amin Allahyar, Carlo Vermeulen, Britta AM Bouwman, Peter HL Krijger, Marjon JAM Verstegen, Geert Geeven, Melissa van Kranenburg, Mark Pieterse, Roy Straver, Judith HI Haarhuis, et al. Enhancer hubs and loop collisions identified from single-allele topologies. *Nature genetics*, 50(8):1151–1160, 2018.
- [16] A Marieke Oudelaar, James OJ Davies, Lars LP Hanssen, Jelena M Telenius, Ron Schwessinger, Yu Liu, Jill M Brown, Damien J Downes, Andrea M Chiariello, Simona Bianco, et al. Single-allele chromatin interactions identify regulatory hubs in dynamic compartmentalized domains. *Nature genetics*, 50(12):1744–1751, 2018.
- [17] Lei Liu, Bokai Zhang, and Changbong Hyeon. Extracting multi-way chromatin contacts from hi-c data. *PLoS Computational Biology*, 17(12):e1009669, 2021.
- [18] Sofia A Quinodoz, Noah Ollikainen, Barbara Tabak, Ali Palla, Jan Marten Schmidt, Elizabeth Detmar, Mason M Lai, Alexander A Shishkin, Prashant Bhat, Yodai Takei, et al. Higher-order inter-chromosomal hubs shape 3d genome organization in the nucleus. *Cell*, 174(3):744–757, 2018.
- [19] Meizhen Zheng, Simon Zhongyuan Tian, Daniel Capurso, Minji Kim, Rahul Maurya, Byoungkoo Lee, Emaly Piecuch, Liang Gong, Jacqueline Jufen Zhu, Zhihui Li, et al. Multiplex chromatin interactions with single-molecule precision. *Nature*, 566(7745):558–562, 2019.

- [20] Aditya S Deshpande, Netha Ulahannan, Matthew Pendleton, Xiaoguang Dai, Lynn Ly, Julie M Behr, Stefan Schwenk, Will Liao, Michael A Augello, Carly Tyer, et al. Identifying synergistic high-order 3d chromatin conformations from genome-scale nanopore concatemer sequencing. *Nature Biotechnology*, pages 1–12, 2022.
- [21] Claire Donnat and Susan Holmes. Tracking network dynamics: A survey of distances and similarity metrics. *arXiv preprint arXiv:1801.07351*, 2018.
- [22] Daniel Zelazo, Mehran Mesbahi, and Mohamed-Ali Belabbas. Graph theory in systems and controls. In *2018 IEEE Conference on Decision and Control (CDC)*, pages 6168–6179. IEEE, 2018.

Reviewers' Comments:

Reviewer #3:

Remarks to the Author:

The major concern for this manuscript is the hypergraph based Laplacian matrix.

First, the specially-designed normalized Laplacian matrix $L = D - HE^{-1}H^T$ or $L = I - D^{-1/2}HE^{-1}H^T D^{-1/2}$ is different from the traditional Laplacian matrices! It is easy to see this by a simple example. For instance, we can consider a hypergraph with two hyperedges (v_1, v_2, v_3) and (v_3, v_4) . (The author can add this example in their paper to demonstrate the difference). The specially defined normalized Laplacian matrix is different from traditional ones (from decomposition of hypergraph into graph).

Second, the special normalized Laplacian matrix is indeed well defined. It has the general Laplacian matrix properties, such as semi-positive definite, smallest eigenvalue is zero, etc.

Third, mathematically, the specially defined normalized Laplacian is the normalized Laplacian with a special weight. More specially, the weight is from order of hyperedges, i.e., matrix E . Note that, in traditional normalized Laplacian matrix, the normalized matrix is obtained from the adjacent matrix (or Laplacian matrix) by weighting over the node degrees. In this paper, the specially defined normalized Laplacian is normalized by both node degrees and hyperedge orders.

Four, the specially designed normalized Laplacian matrix characterizes "pairwise interactions". However, since the interactions are weighted by "the order of hyperedge", it contains certain "multi-way" components.

Five, mathematically, both matrices of HH^T and H^TH are correct and meaningful. Note that H matrix can be viewed as a certain boundary matrix. In this way, HH^T characterizes node-wise interactions and H^TH are (hyper)edge-wise interactions.

Six, more interesting "higher-order" or "multi-way" interactions can be obtained through simplicial complex and hypergraph tensor models. A clique complex can be obtained from hypergraph and higher order Hodge (or combinatorial) Laplacian matrices can be generated accordingly. Hypergraph tensor matrices are also important tool to characterize higher-order interactions.

Minor concern,

The notation of E is improper. In Page 15, E is "the hyperedge set". In Page 16, E is "diagonal matrix".

Point-by-point Response to NCOMMS-21-35046C-Z

Title: Deciphering Multi-way Interactions in the Human Genome

We thank Senior Editor Dr. Cara Eldridge for her time on this manuscript, the editorial team for their input, and the reviewers for their time and valuable comments. Please see the following reviewer comments (in black) and our point-by-point responses (in blue). Based on the reviewers' suggestions, we have revised our manuscript, improved the clarity of our findings, and addressed errors throughout the text.

Reviewer #3 (Remarks to the Author):

The major concern for this manuscript is the hypergraph based Laplacian matrix.

First, the specially-designed normalized Laplacian matrix $L = D - HE^{-1}H^T$ or $L = I - D^{-1/2}HE^{-1}H^T-1/2$ is different from the traditional Laplacian matrices! It is easy to see this by a simple example. For instance, we can consider a hypergraph with two hyperedges (v_1, v_2, v_3) and (v_3, v_4) . (The author can add this example in their paper to demonstrate the difference). The specially defined normalized Laplacian matrix is different from traditional ones (from decomposition of hypergraph into graph).

We thank the Reviewer for this valuable suggestion. We have added the simple hypergraph example mentioned by the Reviewer to the manuscript to that we can clearly demonstrate for readers the difference between the hypergraph Laplacian matrices and the traditional Laplacian matrices.

Second, the special normalized Laplacian matrix is indeed well defined. It has the general Laplacian matrix properties, such as semi-positive definite, smallest eigenvalue is zero, etc.

We thank the Reviewer for this valuable comment.

Third, mathematically, the specially defined normalized Laplacian is the normalized Laplacian with a special weight. More specially, the weight is from order of hyperedges, i.e., matrix E . Note that, in traditional normalized Laplacian matrix, the normalized matrix is obtained from the adjacent matrix (or Laplacian matrix) by weighting over the node degrees. In this paper, the specially defined normalized Laplacian is normalized by both node degrees and hyperedge orders.

We thank the Reviewer for this valuable comment.

Four, the specially designed normalized Laplacian matrix characterizes "pairwise interactions". However, since the interactions are weighted by "the order of hyperedge", it contains certain "multi-way" components.

We thank the Reviewer for this valuable comment.

Five, mathematically, both matrices of HH^T and H^TH are correct and meaningful. Note that H matrix can be viewed as a certain boundary matrix. In this way, HH^T characterizes node-wise interactions and H^TH are (hyper)edge-wise interactions.

We are grateful to the Reviewer for these valuable comments and pointing out the mathematical soundness of our methods.

Six, more interesting "higher-order" or "multi-way" interactions can be obtained through simplicial complex and hypergraph tensor models. A clique complex can be obtained from hypergraph and higher order Hodge (or combinatorial) Laplacian matrices can be generated accordingly. Hypergraph tensor matrices are also important tool to characterize higher-order interactions.

We thank the Reviewer for this valuable comment.

Minor concern,

The notation of E is improper. In Page 15, E is "the hyperedge set". In Page 16, E is "diagonal matrix".

We thank the Reviewer for pointing out this error. We have used the calligraphic font to represent the node set (i.e., \mathcal{V}) and the hyperedge set (i.e., \mathcal{E}) of a hypergraph to avoid the abuse of notations.